# ModernTCN Revisited: A Critical Look at the Experimental Setup in General Time Series Analysis

**Önder Akaçık**                                                    *o.akacik@vu.nl*
*Vrije Universiteit Amsterdam*

**Mark Hoogendoorn**                                               *m.hoogendoorn@vu.nl*
*Vrije Universiteit Amsterdam*

**Reviewed on OpenReview:** *https://openreview.net/forum?id=R2OkKdWmVZ*

## Abstract

While numerous time series models claim state-of-the-art performance, their evaluation often relies on flawed experimental setups, leading to questionable conclusions. This study provides a critical re-evaluation of this landscape, using ModernTCN as a case study. We conduct a rigorous and extended benchmark, correcting methodological issues related to data loading, validation, and evaluation methods, and show that performance claims are sensitive to these details. Additionally, we find that ModernTCN overlooks a line of research in global convolutional models, and our comparison reveals that despite claims of an enlarged effective receptive field (ERF), it falls short of these methods. More than a critique, we introduce an architectural innovation: by embedding irregularly sampled data with a continuous kernel convolution and processing it with the ModernTCN backbone, we achieve new state-of-the-art performance on the challenging PhysioNet 2019 dataset. This work not only provides a robust reassessment of ModernTCN but also serves as an audit of the commonly used general time series analysis experimental setup, which includes tasks such as forecasting, imputation, classification, and anomaly detection.[1]

## 1 Introduction

Time series analysis is a fundamental problem with broad applications across various domains, including weather forecasting (Bi et al., 2023), anomaly detection in spacecraft monitoring (Su et al., 2019b), medical symptom classification (Kiyasseh et al., 2021), and missing data imputation (Luo et al., 2018).

ModernTCN is a general time series analysis model designed to enhance performance across five key time series tasks: long-term forecasting, short-term forecasting, classification, imputation, and anomaly detection. It modernizes traditional Temporal Convolutional Networks (TCNs) Bai et al. (2018) by enlarging the effective receptive field (ERF), drawing inspiration from computer vision advances (Liu et al., 2022a; Ding et al., 2022a) and Transformer architectures (Vaswani et al., 2017). The model also incorporates techniques from transformer-based time series models, such as patching and variable independence approaches (Nie et al., 2023).

The concept of a general time series analysis model, as seen in ModernTCN, builds on the experimental setup introduced by TimesNet (Wu et al., 2023). However, this setup has several flaws. For example, the setup often accelerates testing by splitting data into batches and discarding the last incomplete batch of the test set, which can lead to unfair comparisons and misleading results (Qiu et al., 2024). In anomaly detection, this setup employs an evaluation method prior to scoring, which can lead to an overestimation of performance, as demonstrated by Kim et al. (2022). Additionally, for short-term forecasting and classification, the test set is used for validation, leading to overly optimistic results and an unfair comparison with earlier baselines

---

[1]The official repository for this work is available at: https://github.com/onderakacik/RE-ModernTCN

that adhere to stricter evaluation protocols. Alongside ModernTCN, many papers, such as GPT4TS (Zhou et al., 2023), UniTS (Gao et al., 2024), and TimeMixer++ (Wang et al., 2025), follow this general time series analysis setup.

The primary purpose of this study is to validate this setup using ModernTCN as a means. Furthermore, we extend the benchmarks and experiments to rectify the limitations of the original setup, thereby improving the robustness. Additionally, we address the oversight of global convolutional-based models (Romero et al., 2021b; Gu et al., 2021; Romero et al., 2021a; Knigge et al., 2023) in the original study. By harnessing the synergy between a global convolutional method (Romero et al., 2021b) and ModernTCN, we enhance the model performance on irregularly sampled data.

The contributions of this study are:

- Auditing the general time series analysis experimental setup using ModernTCN as a case study, identifying and correcting critical flaws in data handling and evaluation protocols.

- Innovating ModernTCN by integrating it with a global convolutional method, resulting in a superior model for irregularly sampled data and achieving state-of-the-art performance on the PhysioNet 2019 dataset (Reyna et al., 2019).

- Validating ModernTCN's claims of an extended effective receptive field by visualizing and comparing its ERF against global convolutional methods.

## 2 Related Work

General time series models like ModernTCN, TimesNet and UniTS aim to provide comprehensive solutions across various time series tasks. TimesNet introduces a unified framework for time series analysis by transforming 1D time series into 2D tensors, allowing for the modeling of complex temporal variations using 2D kernels (Wu et al., 2023). UniTS employs task tokenization to integrate predictive and generative tasks into a single framework, using a modified transformer block to capture universal time series representations, enabling transferability across diverse datasets and tasks (Gao et al., 2024). All these models utilize an experimental setup that originates from TimesNet, similar to ModernTCN.

Recent studies are addressing the faulty setups in long-term forecasting and anomaly detection, which are part of the general time series analysis setup. The "Drop Last Trick," as highlighted by Qiu et al. (2024), involves discarding the last batch if it contains fewer instances than the batch size, which can lead to misleading results. Kim et al. (2022) have shown the misleading nature of point adjustment in anomaly detection. Additionally, Liu & Paparrizos (2024) proposes a comprehensive benchmark addressing the known but often ignored issues in the domain of time series anomaly detection.

Transformers have shown strong performance in time series forecasting, as demonstrated by models like PatchTST (Nie et al., 2023). However, MLP-based models have gained popularity by questioning the necessity of transformers, offering similar performance with greater efficiency, as seen in DLinear (Zeng et al., 2023). Convolution-based models, on the other hand, strike a balance by being more efficient than transformers with subquadratic time complexity and more expressive than MLPs (Luo & Wang, 2024; Wang et al., 2022).

In addition to TimesNet and ModernTCN, convolutional models like TCN (Bai et al., 2018) and MICN (Wang et al., 2022) offer distinct approaches to time series analysis. TCN, as described by Bai et al. (2018), employs causal convolutions to ensure no information leakage from future to past, allowing it to handle sequences of any length and map them to output sequences of the same length. This architecture is designed to combine simplicity with autoregressive prediction and long memory, making it effective for sequence modeling. On the other hand, MICN goes beyond causal convolution by proposing a multi-scale convolution structure that combines local features and global correlations in time series, enhancing its ability to capture complex temporal patterns.

In addition to the task-specific models, a significant recent development is the emergence of Time Series Foundation Models (TSFMs), which are pre-trained on massive data to achieve zero-shot forecasting capa-

bilities (Gao et al., 2024; Das et al., 2024; Ansari et al., 2024; Woo et al., 2024; Liu et al., 2024; Auer et al., 2025). These models offer enhanced generalizability and simplified application, often leveraging in-context learning from their extensive pre-training.

Lastly, ModernTCN is inspired by computer vision techniques to enlarge the ERF, but it overlooks a line of research involving models with global convolution capabilities, such as CKConv Romero et al. (2021b), FlexConv Romero et al. (2021a), CCNN Knigge et al. (2023), and S4 Gu et al. (2021). These models achieve substantially larger ERFs through continuous kernel formulations, enabling arbitrarily large memory horizons while maintaining efficiency.

## 3 Methodology

### 3.1 ModernTCN

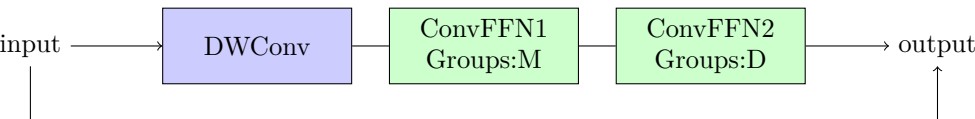

Figure 1: Diagram of the ModernTCN block. Adapted from Luo & Wang (2024).

ModernTCN (Luo & Wang, 2024) introduces a convolutional approach with large kernel sizes inspired by advances in computer vision (Liu et al., 2022c; Ding et al., 2022b; Liu et al., 2022b). It leverages modern convolution techniques, incorporating depthwise and pointwise convolution layers organized similarly to Transformer blocks (Vaswani et al., 2017). The depthwise convolution (DWConv) layer is responsible for learning temporal information among tokens on a per-channel basis, akin to the self-attention module in Transformers. A large kernel is used in DWConv to enhance the effective receptive field, allowing the model to capture long-term dependencies more effectively. The ConvFFN module consists of two pointwise convolution layers that adopt an inverted bottleneck structure. This module is applied twice in the architecture, first along feature dimensions and then between variables, to learn new feature representations.

The pipeline of ModernTCN is as follows:

1. **Input**: The process begins with the input time series $\mathbf{X}_{\text{in}} \in \mathbb{R}^{M \times L}$, where $M$ is the number of variables and $L$ is the input length.

2. **Patchify Embeddings**: The input is transformed into a higher-dimensional space by patching in a variable-independent manner, resulting in $\mathbf{X}_{\text{emb}} \in \mathbb{R}^{M \times D \times N}$, where $D$ is the feature dimension and $N$ is the number of patches.

3. **Backbone**: As shown in Figure 1 each ModernTCN block consists of:
   - **DWConv Layer**: Processes temporal information independently for each variable and each feature, maintaining dimensions $M \times D$.
   - **ConvFFN Module**: This module consists of two pointwise convolution layers and adopts an inverted bottleneck structure. It is applied twice:
     - **ConvFFN1**: Processes each variable independently with groups=M, learning new feature representations independently for each variable. Until this point, the entire pipeline maintains variable independence.
     - **ConvFFN2**: Processes across variables with groups=D, capturing dependencies between different variables. This is the first component in the pipeline that allows cross-variable information mixing.

   These blocks are stacked and organized in a residual manner as shown in Figure 1.

4. **Flatten and Prediction Head**:

- **Forecasting**: The output is reshaped and passed through a prediction head to produce the final time series output $\hat{\mathbf{X}} \in \mathbb{R}^{M \times T}$, where $T$ is the prediction length.
- **Imputation and Anomaly Detection**: The prediction head produces output $\hat{\mathbf{X}} \in \mathbb{R}^{M \times L}$.
- **Classification**: The representation is flattened to and passed through a projection layer with a SoftMax activation to yield the classification result $\hat{\mathbf{X}} \in \mathbb{R}^{1 \times \text{Cls}}$, where Cls is the number of classes.

5. **RevIN**: For all tasks except classification, Stationary Technique RevIN (Kim et al., 2021) is applied. It normalizes the input time series per variable with zero mean and unit standard deviation before patching and embedding. After the forward process, the mean and deviation are added back to the final prediction per variable.

This architecture leverages the decoupling of temporal, feature, and variable dimensions to improve both performance and efficiency in time series analysis, while the enlarged kernel size in DWConv enhances its ability to capture long-term dependencies.

## 3.2 Tasks and Datasets

Table 1: Dataset descriptions. The dataset size is organized in (train, validation, test). The column length refers to prediction length for forecasting tasks, input length for imputation, series length for classification, and sliding window length for anomaly detection. Partially adapted from Wu et al. (2023).

| Tasks | Dataset | Dim(+Static) | Length | Dataset Size | Information (Frequency) |
|---|---|---|---|---|---|
| Long-term forecasting | ETTm1, ETTm2 | 7 | {96, 192, 336, 720} | (34465, 11521, 11521) | Electricity (15 mins) |
| | ETTh1, ETTh2 | 7 | {96, 192, 336, 720} | (8545, 2881, 2881) | Electricity (hourly) |
| | Electricity | 321 | {96, 192, 336, 720} | (18317, 2633, 5261) | Electricity (hourly) |
| | Traffic | 862 | {96, 192, 336, 720} | (12185, 1757, 3509) | Transportation (hourly) |
| | Weather | 21 | {96, 192, 336, 720} | (36792, 5271, 10540) | Weather (10 mins) |
| | Exchange | 8 | {96, 192, 336, 720} | (5120, 665, 1422) | Exchange rate (daily) |
| | ILI | 7 | {24, 36, 48, 60} | (617, 74, 170) | Illness (weekly) |
| Short-term forecasting original | M4-Yearly | 1 | 6 | (23000, 0, 23000) | Demographic |
| | M4-Quarterly | 1 | 8 | (24000, 0, 24000) | Finance |
| | M4-Monthly | 1 | 18 | (48000, 0, 48000) | Industry |
| | M4-Weekly | 1 | 13 | (359, 0, 359) | Macro |
| | M4-Daily | 1 | 14 | (4227, 0, 4227) | Micro |
| | M4-Hourly | 1 | 48 | (414, 0, 414) | Other |
| Extended | ETTm1 | 7 | {6, 12, 24} | (34465, 11521, 11521) | Electricity (15 mins) |
| | ETTh1 | 7 | {6, 12, 24} | (8545, 2881, 2881) | Electricity (hourly) |
| Imputation | ETTm1, ETTm2 | 7 | 96 | (34465, 11521, 11521) | Electricity (15 mins) |
| | ETTh1, ETTh2 | 7 | 96 | (8545, 2881, 2881) | Electricity (15 mins) |
| | Electricity | 321 | 96 | (18317, 2633, 5261) | Electricity (15 mins) |
| | Weather | 21 | 96 | (36792, 5271, 10540) | Weather (10 mins) |
| Classification original | EthanolConcentration | 3 | 1751 | (261, 0, 263) | Alcohol industry |
| | FaceDetection | 144 | 62 | (5890, 0, 3524) | Face (250Hz) |
| | Handwriting | 3 | 152 | (150, 0, 850) | Handwriting |
| | Heartbeat | 61 | 405 | (204, 0, 205) | Heart beat |
| | JapaneseVowels | 12 | 29 | (270, 0, 370) | Voice |
| | PEMS-SF | 963 | 144 | (267, 0, 173) | Transportation (daily) |
| | SelfRegulationSCP1 | 6 | 896 | (268, 0, 293) | Healthcare (256Hz) |
| | SelfRegulationSCP2 | 7 | 1152 | (200, 0, 180) | Healthcare (256Hz) |
| | SpokenArabicDigits | 13 | 93 | (6599, 0, 2199) | Voice (11025Hz) |
| | UWaveGestureLibrary | 3 | 315 | (120, 0, 320) | Gesture |
| Extended | Speech Commands MFCC | 20 | 161 | (24483, 5246, 5246) | Voice(16000Hz) |
| | Speech Commands Raw | 1 | 16000 | (24483, 5246, 5246) | Voice(16000Hz) |
| | PhysioNet | 34(+5) | 72 | (28235, 6050, 6050) | Healthcare |
| | Character Trajectories | 3 | [60-182] | (1209, 213, 1436) | Handwriting |
| Anomaly detection | SMD | 38 | 100 | (566724, 141681, 708420) | Server machine |
| | MSL | 55 | 100 | (44653, 11664, 73729) | Spacecraft |
| | SMAP | 25 | 100 | (108146, 27037, 427617) | Spacecraft |
| | SWaT | 51 | 100 | (396000, 99000, 449919) | Infrastructure |
| | PSM | 25 | 100 | (105984, 26497, 87841) | Server machine |
| Extended | TSB-AD-U | 1 | It is curated from 23 different datasets. See Appendix A.2 for details. | | |
| | TSB-AD-M | [2-248] | It is curated from 17 different datasets. See Appendix A.2 for details. | | |

Time series analysis encompasses a variety of tasks, each designed to extract different insights from sequential data. This study focuses on five primary tasks: long-term forecasting, short-term forecasting, imputation, classification, and anomaly detection. An overview of all the datasets used is given in Table 1.

**Long-term forecasting** involves predicting future values over an extended time horizon, often requiring the model to capture complex temporal dependencies. ModernTCN (Luo & Wang, 2024) uses datasets such as ETT (Zhou et al., 2021), Electricity (UCI), Traffic (PeM), Weather (Wet), Exchange (Lai et al., 2018),

and ILI (CDC), covering real-world applications from different domains. In these multivariate forecasting tasks, the model predicts future values for all variables in the time series simultaneously.

**Short-term forecasting** focuses on predicting values over a shorter time frame, typically requiring the model to capture immediate trends and patterns. ModernTCN uses the M4 dataset (Makridakis, 2018), which contains univariate time series with different frequencies from different domains. We extend the experiments to ETT (Zhou et al., 2021) by adapting it for short term multivariate forecasting.

**Imputation** aims to fill in missing values in a time series, which is crucial for handling incomplete data and ensuring data quality. ModernTCN (Luo & Wang, 2024) uses ETT (Zhou et al., 2021), Electricity (UCI), and Weather (Wet) datasets for this task.

**Classification** is sequence-level that verifies the model's capacity in high-level representation learning. ModernTCN (Luo & Wang, 2024) uses 10 multivariate datasets from the UEA Time Series Classification Archive (Bagnall et al., 2018). We extend to Speech Commands (Warden, 2018) and PhysioNet (Reyna et al., 2019), following the experimental setup from CKConv (Romero et al., 2021b). Speech Commands enables us to compare ModernTCN with other convolution-based models that has global receptive fields. PhysioNet's irregular sampling and high proportion of missing values present a challenging classification task. Additionally, we extend the evaluation to variable length time series classification using Character Trajectories dataset (Bagnall et al., 2018).

**Anomaly detection** seeks to identify unusual patterns or outliers in a time series, which is essential for detecting abnormal events and potential issues. ModernTCN (Luo & Wang, 2024) evaluates anomaly detection performance on SMD (Su et al., 2019a), MSL (Hundman et al., 2018), SMAP (Hundman et al., 2018), SWaT (Mathur & Tippenhauer, 2016), and PSM (Abdulaal et al., 2021), covering service monitoring, space & earth exploration, and water treatment applications. We extend the evaluation to TSB-AD (Liu & Paparrizos, 2024), a comprehensive benchmark for anomaly detection consisting of 1070 high-quality time series from a diverse collection of 40 datasets (see Table 19 for details).

### 3.3 Experimental Setup

For the reproduction of the experiments from ModernTCN, the optimal settings specified in the paper (Luo & Wang, 2024) and the source code[2] are used. For the extended experiments we conducted a grid search over the relevant hyperparameters for each setup (See Appendix A.1 for details). For the best-performing hyperparameters, we reported the mean and standard deviation over 5 runs.

Upon investigating the source code, we identified issues affecting the five tasks used in Luo & Wang (2024). We fixed the errors and rerun the experiment for long-term forecasting and imputation tasks. For short-term forecasting, classification and anomaly detection, we extended the experiments to new datasets instead.

#### 3.3.1 Issues in the Original Setup

**Drop Last Trick.** As pointed out by Qiu et al. (2024), many implementations of existing methods often employ a "Drop Last Trick" during the testing phase (Nie et al., 2022; Wang et al., 2022; Zhou et al., 2022; 2021), which involves discarding the last batch if it contains fewer instances than the batch size. This approach may lead to unfair comparisons by excluding part of the test data. Upon reviewing the source code, we found this trick is employed for long-term forecasting and imputation tasks. Given that most datasets in the long-term forecasting experiments use a batch size of 512, this could result in discarding a significant amount of the test set (See Appendix 25 for further analysis). Therefore, we additionally experiment without employing this trick to assess its impact on the results and ensure a fair comparison (Table 2 and Table 4).

**Data Leakage from Test to Validation.** As detailed in Table 1, the M4 dataset (Makridakis, 2018) for short-term forecasting and the UEA dataset (Bagnall et al., 2018) for classification are provided with only train and test splits. Upon reviewing the source code, we identify that the test set is used for validation. This practice introduces an information leakage as the model gains indirect exposure to the test data, leading to

---

[2] The official source code for ModernTCN can be found at `https://github.com/luodhhh/ModernTCN`

an overestimation of its generalization capabilities. Consequently, the reproduction results for these datasets are excluded from the main results section and are instead presented in Appendix E.1 and Appendix E.2.

**Point Adjustment in Anomaly Detection.** Reconstruction-based anomaly detection identifies anomalies when reconstruction errors between input sequences and their model-generated reconstructions exceed a threshold. The threshold is calculated by taking a specific percentile of the reconstruction error scores from the training data, ensuring that a defined percentage of the data is considered normal. But ModernTCN calculates the threshold using both training and test sets, introducing information leakage. We reproduce the experiments using thresholds calculated from training data only. More importantly, the evaluation employs "point adjustment" (PA), which considers an entire anomaly segment correctly detected if just one point within it is flagged - a protocol that Kim et al. (2022) demonstrated can severely overestimate performance. To illustrate this overestimation, we implemented a naive baseline that periodically flags every 100th time point as anomalous, which should perform well under PA despite having no actual anomaly detection capability. 100 is set as the period in the naive approach because the sliding window size is 100.

### 3.3.2 Extended Setups

**Extending Short-term Forecasting to ETT.** Inspired by the appendices in Luo & Wang (2024), we evaluate short-term multi-variate forecasting on the ETT dataset (Zhou et al., 2021). While ModernTCN (Luo & Wang, 2024) keeps the input sequence length constant at 2x the prediction length, we explored the input sequence length as a hyperparameter, experimenting with 2x, 3x, and 4x the prediction length. We consider three prediction lengths: 6, 12, and 18 time points, selecting the best-performing input length for reporting. We did a hyperparameter search for the kernel size and employed the optimal hyperparameters from the corresponding long-term forecasting experiments for the rest. For comparison, we decided to include three top-performing models of different kinds from Qiu et al. (2024): TimesNet (convolution-based) (Wu et al., 2023), PatchTST (transformer-based) (Nie et al., 2023), and DLinear (MLP-based) (Zeng et al., 2023).

**Extending Classification to Speech Commands.** For the classification task, we employ the Speech Commands dataset (Warden, 2018). This dataset has two versions: Speech Commands Raw and Speech Commands MFCC. The Speech Commands Raw version consists of 105,809 one-second audio recordings of 35 spoken words sampled at 16kHz. Following the preprocessing steps of Kidger et al. (2020), we extract 34,975 recordings from ten spoken words to construct a balanced classification problem. The Speech Commands MFCC utilizes mel-frequency cepstrum coefficients extracted from the raw data, resulting in time series of length 161 and 20 channels. This dataset is frequently used by convolution-based models with global receptive fields (Romero et al., 2021b;a; Knigge et al., 2023; Gu et al., 2021).

**Extending to Variable Length Time Series Classification.** The CharacterTrajectories dataset is part of the UEA time series classification archive (Bagnall et al., 2018). It consists of 2,858 time series of varying lengths, each with 3 channels representing the x, y positions and the pen tip force while writing a Latin alphabet character in a single stroke. The goal is to classify which of the 20 different characters was written using the time series data. The length of the time series varies between 60 and 182. Unlike Romero et al. (2021b), we retained the original test set and created a validation split by taking 15% of the training set. This approach ensures comparability with Lee & Shin (2025). Additionally, we reran the CKConv (Romero et al., 2021b) experiments with this new setup using its official codebase[3].

**Extending to Irregularly Sampled Data.** The PhysioNet 2019 challenge on sepsis prediction provides an irregularly sampled, partially observed dataset with 40,335 time series of variable lengths, capturing ICU patient data (Reyna et al., 2019). Each series includes 5 static features, like age, and 34 dynamic features, such as respiration rate, with only 10.3% of values observed. Following Kidger et al. (2020); Romero et al. (2021b), we analyze the first 72 hours of patient data to predict sepsis development over their entire stay, which can last up to a month. This setup tests the model's ability to handle sparse and irregular data, creating a robust test bed for ModernTCN, which claims state-of-the-art performance on the imputation task. Similar to ModernTCN, we ablate the cross-variable component to assess its impact on handling missing values in a multivariate setting. Most importantly, we introduce an innovation in our reproducibility

---

[3]The official CKConv codebase can be found at `https://github.com/dwromero/ckconv`

study by combining a global convolutional method (Romero et al., 2021b) with ModernTCN, as detailed in Table 10.

**Extending Anomaly Detection to TSB-AD.** Extending our anomaly detection experiments to the TSB-AD datasets and benchmark (Liu & Paparrizos, 2024) is beneficial due to its comprehensive nature, encompassing over 40 datasets and a wide array of baselines. The benchmark includes 40 time-series anomaly detection algorithms categorized into statistical, neural network-based, and foundation model-based methods. This extension addresses critical issues in the field, such as flawed datasets and biased evaluation measures, by employing VUS-PR (Volume Under the Surface of Precision-Recall) as a more reliable evaluation metric. VUS-PR is threshold-independent, summarizing performance across all possible anomaly score thresholds, thus providing a holistic and stable evaluation. It is more robust to temporal misalignments and less biased towards random or noisy scores compared to traditional measures like PA-F1 and AUC-ROC. By incorporating a tolerance buffer around anomaly boundaries, VUS-PR enhances the relevance of anomaly scoring, making it a superior choice for evaluating time-series anomaly detection models.

# 4 Experimental Results

## 4.1 Results Reproducing the Original Paper

### 4.1.1 Long-Term Forecasting

The "Drop Last Trick" significantly impacts datasets like ETTh and Exchange, where a substantial percentage of the test set is discarded, as in Table 2. This leads to notable changes in MSE/MAE, particularly for these datasets, as a high percentage of data is dropped. Additionally, the ILI dataset exhibits significant changes in MSE/MAE, likely due to its small size (Table 1). For further analysis, please refer to the Appendix D.

All results in Table 3 are presented after the "Drop Last Trick" is corrected. PatchTST (Nie et al., 2023), TimesNet (Wu et al., 2023), and DLinear (Zeng et al., 2023) are the best-performing methods based on transformer, convolution, and MLP architectures, respectively, as identified by Qiu et al. (2024). We include MICN (Wang et al., 2022) and TCN (Bai et al., 2018) as extra convolution-based baselines. Additionally, we include results for several Time Series Foundation Models (TSFMs) in a zero-shot setting (Gao et al., 2024; Das et al., 2024; Ansari et al., 2024; Woo et al., 2024; Liu et al., 2024; Auer et al., 2025).

The ranks in Table 3 are calculated independently for task-specific models and foundational models. While PatchTST is the top-performing task-specific model, ModernTCN remains highly competitive by frequently ranking second and outperforming other convolutional baselines. Notably, TiRex not only leads the foundational models but also rivals the best task-specific models with superior MAE in most cases and competitive MSE, even though only the Electricity dataset from our benchmark was part of its pre-training data (Auer et al., 2025).

Table 2: Long-term forecasting reproduced results of ModernTCN. Reported, Rerun and Fixed refer to original paper results, results before and after the "Drop Last Trick" is fixed respectively. Dropped % shows percentage of test data points excluded when using the trick. Δ MSE/MAE shows performance change when the trick is fixed and it is calculated as "Δ = Fixed - Rerun". Results are averaged from 4 different prediction lengths: {24, 36, 48, 60} for ILI and {96, 192, 336, 720} for others.

| Result Type | ETTh1 MSE | ETTh1 MAE | ETTh2 MSE | ETTh2 MAE | ETTm1 MSE | ETTm1 MAE | ETTm2 MSE | ETTm2 MAE | Electricity MSE | Electricity MAE | Weather MSE | Weather MAE | Traffic MSE | Traffic MAE | Exchange MSE | Exchange MAE | ILI MSE | ILI MAE |
|---|---|---|---|---|---|---|---|---|---|---|---|---|---|---|---|---|---|---|
| Reported | 0.404 | 0.420 | 0.322 | 0.379 | 0.351 | 0.380 | 0.253 | 0.314 | 0.156 | 0.253 | 0.224 | 0.264 | 0.396 | 0.270 | 0.302 | 0.366 | 1.440 | 0.786 |
| Rerun | 0.404 | 0.421 | 0.322 | 0.379 | 0.355 | 0.383 | 0.255 | 0.319 | 0.159 | 0.256 | 0.224 | 0.267 | 0.401 | 0.274 | 0.305 | 0.368 | 1.490 | 0.797 |
| Fixed | 0.419 | 0.429 | 0.347 | 0.394 | 0.356 | 0.383 | 0.254 | 0.318 | 0.163 | 0.259 | 0.226 | 0.267 | 0.410 | 0.280 | 0.343 | 0.392 | 2.000 | 0.892 |
| Dropped % | 11.10% | 11.10% | 11.10% | 11.10% | 2.23% | 2.23% | 2.23% | 2.23% | 0.25% | 0.25% | 1.64% | 1.64% | 0.60% | 0.60% | 14.49% | 14.49% | 5.88% | 5.88% |
| Δ MSE/MAE | +0.015 | +0.008 | +0.025 | +0.015 | +0.001 | 0.000 | -0.001 | -0.001 | +0.004 | +0.003 | +0.003 | +0.001 | +0.009 | +0.006 | +0.038 | +0.024 | +0.510 | +0.095 |

### 4.1.2 Imputation

The results presented in Table 4 demonstrate that due to the smaller batch size used in the imputation task, as referenced in Table 25, the percentage of dropped points from the test set is minimal. This minimal drop percentage explains why the results remained unchanged after the "Drop Last Trick" was corrected.

Table 3: Long-term forecasting task results after the "Drop Last Trick" is fixed. All results averaged from 4 different prediction lengths: {24, 36, 48, 60} for ILI and {96, 192, 336, 720} for others. Lower MSE/MAE indicates better performance. Bold indicates best performance, and underline indicates second-best performance. Ranks are calculated independently for task-specific models and foundational models.

| Model | 1st/2nd | ETTh1 | | ETTh2 | | ETTm1 | | ETTm2 | | Electricity | | Weather | | Traffic | | Exchange | | ILI | |
|---|---|---|---|---|---|---|---|---|---|---|---|---|---|---|---|---|---|---|---|
| | | MSE | MAE | MSE | MAE | MSE | MAE | MSE | MAE | MSE | MAE | MSE | MAE | MSE | MAE | MSE | MAE | MSE | MAE |
| ModernTCN | 3/13 | 0.420 | 0.429 | 0.348 | 0.395 | 0.355 | 0.382 | 0.254 | 0.318 | 0.164 | 0.258 | 0.226 | 0.267 | 0.413 | 0.285 | 0.343 | 0.392 | 2.078 | 0.929 |
| PatchTST | 13/3 | 0.411 | 0.428 | 0.347 | 0.389 | 0.349 | 0.381 | 0.255 | 0.313 | 0.163 | 0.261 | 0.225 | 0.262 | 0.405 | 0.283 | 0.352 | 0.397 | 1.770 | 0.859 |
| TimesNet | 0/0 | 0.459 | 0.455 | 0.394 | 0.416 | 0.430 | 0.428 | 0.294 | 0.332 | 0.186 | 0.287 | 0.261 | 0.287 | 0.626 | 0.328 | 0.421 | 0.442 | 2.174 | 0.951 |
| DLinear | 1/1 | 0.420 | 0.432 | 0.492 | 0.478 | 0.354 | 0.377 | 0.259 | 0.324 | 0.167 | 0.264 | 0.239 | 0.290 | 0.434 | 0.295 | 0.349 | 0.411 | 2.185 | 1.040 |
| MICN | 1/1 | 0.420 | 0.447 | 0.482 | 0.471 | 0.355 | 0.383 | 0.294 | 0.357 | 0.179 | 0.290 | 0.239 | 0.289 | 0.539 | 0.313 | 0.320 | 0.396 | 2.368 | 1.049 |
| TCN | 0/0 | 1.004 | 0.787 | 3.398 | 1.545 | 1.190 | 0.840 | 2.465 | 1.177 | 0.385 | 0.448 | 0.409 | 0.422 | 1.061 | 0.556 | 1.785 | 1.093 | 4.320 | 1.359 |
| Chronos | 0/7 | 0.445 | 0.421 | 0.368 | 0.381 | 0.425 | 0.388 | 0.292 | 0.319 | 0.166 | 0.250 | 0.266 | 0.280 | 0.448 | 0.269 | 0.489 | 0.473 | 2.865 | 1.006 |
| TimesFM | 2/6 | 0.451 | 0.437 | 0.403 | 0.411 | 0.429 | 0.416 | 0.335 | 0.346 | 0.155 | 0.245 | 0.232 | 0.251 | 0.370 | 0.244 | 0.433 | 0.446 | 1.812 | 0.849 |
| UniTS | 1/0 | 0.528 | 0.491 | 0.406 | 0.418 | 0.713 | 0.553 | 0.321 | 0.355 | 0.432 | 0.488 | 0.291 | 0.313 | 0.725 | 0.582 | 0.424 | 0.448 | 4.066 | 1.459 |
| MOIRAI | 1/4 | 0.433 | 0.429 | 0.360 | 0.385 | 0.518 | 0.437 | 0.340 | 0.356 | 0.202 | 0.286 | 0.308 | 0.292 | 0.381 | 0.222 | 0.436 | 0.436 | 3.415 | 1.207 |
| Timer | 1/1 | 0.432 | 0.431 | 0.361 | 0.400 | 0.666 | 0.535 | 0.292 | 0.343 | 0.182 | 0.272 | 0.265 | 0.302 | 0.451 | 0.305 | 0.501 | 0.483 | 3.721 | 1.355 |
| TiRex | 14/2 | 0.433 | 0.416 | 0.330 | 0.362 | 0.378 | 0.365 | 0.263 | 0.302 | 0.155 | 0.239 | 0.226 | 0.247 | 0.413 | 0.253 | 0.426 | 0.430 | 1.741 | 0.821 |

In our comparison of ModernTCN with TimesNet, PatchTST, DLinear, and MICN, ModernTCN clearly outperforms all other methods (Table 5). TimesNet ranks second, which aligns with the fact that both ModernTCN and TimesNet are designed for general time series analysis, whereas PatchTST, DLinear, and MICN are specifically developed for time series forecasting. Additionally, PatchTST and DLinear are variable-independent models, while this task is inherently multivariate. The ability to model cross-variable interactions may facilitate the imputation of missing values by leveraging information from existing variables at the same time point, thereby modeling inter-variate dependencies effectively.

Table 4: Imputation task reproduced results of ModernTCN. Reported, Rerun and Fixed refer to original paper results, results before and after the "Drop Last Trick" is fixed respectively. Dropped % shows percentage of test data points excluded when using the trick. $\Delta$ MSE/MAE shows performance change when the trick is fixed and it is calculated as "$\Delta$ = Fixed - Rerun". Results are averaged from 4 different mask ratios {12.5%, 25%, 37.5%, 50%}.

| | ETTh1 | | ETTh2 | | ETTm1 | | ETTm2 | | Electricity | | Weather | |
|---|---|---|---|---|---|---|---|---|---|---|---|---|
| Result Type | MSE | MAE | MSE | MAE | MSE | MAE | MSE | MAE | MSE | MAE | MSE | MAE |
| Reported | 0.050 | 0.150 | 0.042 | 0.131 | 0.020 | 0.093 | 0.019 | 0.082 | 0.073 | 0.187 | 0.027 | 0.044 |
| Rerun | 0.050 | 0.150 | 0.042 | 0.131 | 0.021 | 0.094 | 0.020 | 0.083 | 0.073 | 0.186 | 0.027 | 0.044 |
| Fixed | 0.050 | 0.150 | 0.042 | 0.131 | 0.021 | 0.094 | 0.020 | 0.083 | 0.073 | 0.186 | 0.027 | 0.044 |
| Dropped % | 0.03% | 0.03% | 0.03% | 0.03% | 0.01% | 0.01% | 0.01% | 0.01% | 0.25% | 0.25% | 0.11% | 0.11% |
| $\Delta$ MSE/MAE | 0.000 | 0.000 | 0.000 | 0.000 | 0.000 | 0.000 | 0.000 | 0.000 | 0.000 | 0.000 | 0.000 | 0.000 |

Table 5: Imputation task results after the "Drop Last Trick" is fixed. We randomly mask {12.5%, 25%, 37.5%, 50%} time points in length-96 time series. The results are averaged from 4 different mask ratios. A lower MSE or MAE indicates better performance. Bold indicates best performance, and underline indicates second-best performance.

| | ETTh1 | | ETTh2 | | ETTm1 | | ETTm2 | | Electricity | | Weather | |
|---|---|---|---|---|---|---|---|---|---|---|---|---|
| Model | MSE | MAE | MSE | MAE | MSE | MAE | MSE | MAE | MSE | MAE | MSE | MAE |
| ModernTCN | 0.050 | 0.150 | 0.042 | 0.131 | 0.021 | 0.094 | 0.020 | 0.083 | 0.073 | 0.186 | 0.027 | 0.044 |
| TimesNet | 0.089 | 0.199 | 0.051 | 0.149 | 0.027 | 0.107 | 0.022 | 0.090 | 0.094 | 0.211 | 0.031 | 0.056 |
| PatchTST | 0.134 | 0.238 | 0.066 | 0.164 | 0.045 | 0.133 | 0.028 | 0.098 | 0.091 | 0.209 | 0.033 | 0.057 |
| DLinear | 0.201 | 0.306 | 0.142 | 0.259 | 0.093 | 0.206 | 0.096 | 0.208 | 0.132 | 0.260 | 0.052 | 0.110 |
| MICN | 0.125 | 0.250 | 0.205 | 0.307 | 0.070 | 0.182 | 0.144 | 0.249 | 0.119 | 0.247 | 0.056 | 0.128 |

### 4.1.3 Anomaly Detection

As shown in Table 6, the rerun results generally align with those reported in the original paper. When eliminating data leakage by calculating thresholds using only training data, we observe a slight performance degradation. Most notably, our naive baseline—which simply flags every 100th timepoint as anomalous—outperforms ModernTCN across most datasets, achieving the highest average F1 score. This finding

raises significant questions about the effectiveness of sophisticated models for anomaly detection when evaluated using point adjustment protocols, supporting the critique by Kim et al. (2022) that such evaluation methods can severely overestimate model performance.

Table 6: Anomaly Detection task. P: precision, R: recall, F1: F1-score (all values in %). Reported: scores from Luo & Wang (2024). Rerun: naively reproduced results. No Leakage: results calculating the thresholds using only training data. Naive: baseline approach. Bold: best score between valid models (No Leakage and Naive). Avg.: average F1 score across all 5 datasets.

| Model | SMD | | | MSL | | | SMAP | | | SWaT | | | PSM | | | Avg. |
|---|---|---|---|---|---|---|---|---|---|---|---|---|---|---|---|---|
| | P | R | F1 | P | R | F1 | P | R | F1 | P | R | F1 | P | R | F1 | |
| Reported | 87.86 | 83.85 | 85.81 | 83.94 | 85.93 | 84.92 | 93.17 | 57.69 | 71.26 | 91.83 | 95.98 | 93.86 | 98.09 | 96.38 | 97.23 | 86.62 |
| Rerun | 87.43 | 81.64 | 84.44 | 89.59 | 74.94 | 81.61 | 90.82 | 55.93 | 69.23 | 95.77 | 90.28 | 92.94 | 98.65 | 94.57 | 96.57 | 84.96 |
| No Leakage | 78.42 | 83.76 | 81.00 | 86.07 | 77.12 | 81.35 | 92.31 | 55.42 | 69.26 | **93.45** | 93.16 | 93.30 | **98.62** | **94.58** | **96.56** | 84.29 |
| Naive | **79.86** | **91.40** | **85.24** | **90.14** | **80.76** | **85.19** | **93.27** | **94.48** | **93.87** | 93.05 | **96.93** | **94.95** | 97.32 | 94.55 | 95.91 | **91.03** |

## 4.2 Results Beyond the Original Paper

### 4.2.1 Short-Term Forecasting on ETT

Referring to the results presented in Table 7, ModernTCN exhibits competitive performance, particularly in the ETTm1 dataset, where it achieves the best results for very short-term forecasting. This aligns with findings from Yang et al. (2024), which demonstrated ModernTCN's effectiveness for ultra-short-term multi-step prediction tasks. However, in ETTh1 dataset, PatchTST demonstrates superior performance.

Table 7: Short-Term Forecasting Results on ETT (ETTh1 and ETTm1). 6x1h, 12x1h, and 18x1h refer to steps ahead forecasting for ETTh1, while 6x15m, 12x15m, and 18x15m refer to steps ahead forecasting for ETTm1. Results are averaged over 5 runs, and both mean and standard deviation are reported.

| | 6x15m | | 12x15m | | 18x15m | | 6x1h | | 12x1h | | 18x1h | |
|---|---|---|---|---|---|---|---|---|---|---|---|---|
| Model | MSE | MAE | MSE | MAE | MSE | MAE | MSE | MAE | MSE | MAE | MSE | MAE |
| ModernTCN | **0.127** ± 0.001 | **0.216** ± 0.001 | **0.204** ± 0.001 | **0.269** ± 0.001 | **0.220** ± 0.001 | **0.296** ± 0.001 | 0.300 ± 0.006 | 0.350 ± 0.003 | 0.300 ± 0.001 | 0.346 ± 0.001 | 0.308 ± 0.001 | 0.351 ± 0.001 |
| PatchTST | 0.138 ± 0.003 | 0.222 ± 0.001 | 0.244 ± 0.004 | 0.295 ± 0.005 | 0.240 ± 0.002 | 0.312 ± 0.001 | **0.253** ± 0.003 | **0.321** ± 0.001 | **0.278** ± 0.003 | **0.333** ± 0.002 | **0.288** ± 0.002 | **0.344** ± 0.002 |
| TimesNet | 0.144 ± 0.002 | 0.235 ± 0.001 | 0.231 ± 0.003 | 0.295 ± 0.003 | 0.256 ± 0.003 | 0.326 ± 0.001 | 0.316 ± 0.002 | 0.367 ± 0.001 | 0.337 ± 0.007 | 0.383 ± 0.004 | 0.348 ± 0.006 | 0.393 ± 0.004 |
| DLinear | 0.146 ± 0.001 | 0.233 ± 0.001 | 0.287 ± 0.001 | 0.323 ± 0.001 | 0.303 ± 0.001 | 0.347 ± 0.001 | 0.284 ± 0.002 | 0.339 ± 0.001 | 0.292 ± 0.001 | 0.341 ± 0.002 | 0.300 ± 0.001 | 0.347 ± 0.001 |

### 4.2.2 Classification on Speech Commands

ModernTCN exhibits limited parameter efficiency, particularly when handling many input variables. Its design maintains a separate feature space for each variable throughout the network, which inflates the model size as the number of variables increases. For instance, in the Speech Commands MFCC dataset with 20 variables and a feature dimension of 32, this results in over 4 million parameters. In contrast, CCNN (Knigge et al., 2023) projects all variables into a shared model space at the input stage, resulting in significantly greater parameter efficiency.

This inefficiency becomes more apparent when comparing across datasets. On the Speech Commands Raw dataset, which contains only a single variable, ModernTCN's selected model size of 512 results in a much smaller model with approximately 500K parameters. Despite the challenge posed by the long input sequence length (16,000), ModernTCN performs better than CKConv in this setting. This is primarily due to two factors: the aggressive downsampling of the raw signal during embedding and the use of a large model dimension (512), which enhances the model's expressiveness. However, even in this favorable setting, ModernTCN still underperforms compared to more advanced global convolutional baselines.

We further analyze ModernTCN's ability to model long-range dependencies through effective receptive field (ERF) visualizations, following the methodology of Kim et al. (2023) and Ding et al. (2022a), as adapted by Luo & Wang (2024). We sample 50 sequences from the Speech Commands MFCC validation set and compute the relative gradient contributions of each input timepoint to the center of the feature map after a fixed number of blocks. The resulting visualizations are shown in Figure 2, where lighter regions indicate stronger influence.

Global models like CCNN{4,140} achieve a much larger ERF with significantly fewer parameters (200K vs. 4M) due to their global kernel design. This enables them to better capture long-range dependencies and contributes to their superior classification performance, as reported in Table 8. Additional implementation details and extended analysis for ERFs are available in Appendix B.

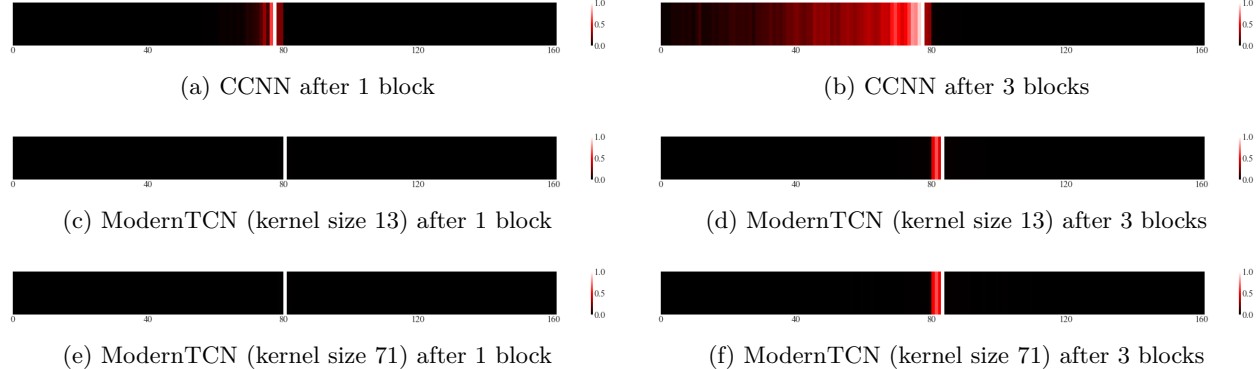

Figure 2: Comparison of effective receptive fields (ERF) for different models. Each row shows a different model configuration, while columns show the ERF after 1 block (left) and 3 blocks (right).

Table 8: Classification Results on Speech Commands - Accuracy

| Model | Size | Speech Commands MFCC | Speech Commands Raw |
|---|---|---|---|
| CKCNN-Seq (Romero et al., 2021b) | 98K | 0.953 | 0.717 |
| S4 (Gu et al., 2021) | 300K | 0.940 | 0.983 |
| FlexTCN-6 (Romero et al., 2021a) | 375K | 0.977 | 0.917 |
| CCNN{4,140} (Knigge et al., 2023) | 200K | 0.950 | 0.983 |
| CCNN{6,380} (Knigge et al., 2023) | 2M | **0.980** | **0.984** |
| ModernTCN | 4M/500k | $0.922 \pm 0.001$ | $0.843 \pm 0.007$ |

### 4.2.3 Variable Length Time Series Classification

Both ModernTCN and CKConv demonstrate strong performance in the variable length time series classification task (Table 9), outperforming the best baselines from Lee & Shin (2025). The high scores achieved by these models can be attributed to the nature of the task. In this context, convolutional models like ModernTCN and CKConv benefit from their ability to extract features through template matching as they slide over the time dimension. CKConv, in particular, excels in this task, likely due to the continuous nature of the handwriting trajectories. By representing the 1D convolutional kernel as a continuous function, CKConv effectively captures the underlying patterns in the data, leading to its exceptional performance.

Table 9: Results of Variable Length Time Series Classification (Accuracy and MAP) over 5 runs, with values representing the mean and standard deviation. ModernTCN and CKConv were adapted, while the other baselines were selected from (Lee & Shin, 2025) as the best variants/models for both metrics.

| Model | Accuracy (%) | MAP (%) |
|---|---|---|
| ModernTCN | $95.02 \pm 0.9$ | $94.42 \pm 0.9$ |
| CKConv | $99.07 \pm 0.3$ | $99.00 \pm 0.3$ |
| Refined TIP | $78.11 \pm 1.18$ | $83.35 \pm 0.83$ |
| InceptionTime (IP) | $82.81 \pm 0.40$ | $69.28 \pm 0.45$ |

### 4.2.4 Irregularly Sampled Time Series Classification

Sepsis prediction is a critical task, as early detection significantly improves patient outcomes. The PhysioNet Sepsis Prediction dataset (Reyna et al., 2019) presents a challenging test bed due to its irregularly sampled nature, which results in missing values. CKConv demonstrates strong performance on this task, surpassing models specifically designed for irregularly sampled data (Kidger et al., 2020).

The ability of ModernTCN to handle missing values is evidenced by its superior performance in the imputation tasks within this study. This capability extends to irregularly sampled data, as shown in Table 10. Additionally, ModernTCN's proficiency in mixing information between variables through its cross-variable component (ConvFFN2) is crucial, as its removal leads to a performance drop. This is indicated as "ModernTCN - Variable Independent," where the model becomes variable-independent until the fully connected projection layer at the end.

Both ModernTCN and CKConv exhibit strong performance on data with missing values. To harness their synergistic potential, we combined them by employing CKConv to embed the raw signals within ModernTCN, while retaining the ModernTCN pipeline for subsequent processing. This approach, denoted as "CKConv + ModernTCN," outperforms all other baselines. The improvement is statistically significant in both comparisons—against CKConv and ModernTCN alone (Wilcoxon signed-rank test, one-sided $p = 0.031$ for both). This combination leverages two key properties of CKConv: (1) the continuous nature of the kernel, which inherently handles irregularly sampled data, and (2) parameter efficiency, as the parameter size of the continuous kernel is decoupled from the input sequence length. This innovative combination could serve as a promising starting point for future studies, particularly in testing on irregularly sampled datasets, including those with very long sequence lengths.

Table 10: Comparison of Model Performance on the PhysioNet Dataset (AUC $\pm$ Std Dev)

| Model | AUC $\pm$ Std Dev |
|---|---|
| CKConv | $0.883 \pm 0.011$ |
| ModernTCN | $0.895 \pm 0.007$ |
| ModernTCN - Variable Independent | $0.875 \pm 0.004$ |
| CKConv + ModernTCN | $0.904 \pm 0.004$ |

### 4.2.5 Anomaly Detection

In the context of anomaly detection, ModernTCN's performance varies significantly between univariate and multivariate time series datasets. On the univariate TSB-AD-U dataset (Figure 3(a)), ModernTCN ranks among the lowest-performing methods, placing third to last. This result aligns with the findings of Liu & Paparrizos (2024), which suggest that statistical methods often provide more effective and robust solutions for univariate time-series anomaly detection. Conversely, on the multivariate TSB-AD-M dataset (Figure 3(b)), ModernTCN emerges as the best-performing method. This success can be attributed to the extensive engineering and design considerations within ModernTCN that specifically address the complexities of multivariate time series data.

## 5 Discussion

This study provides a critical re-evaluation of ModernTCN (Luo & Wang, 2024), using it as a case study to audit the general time series analysis experimental setup. Our rigorous and extended benchmark revealed that performance claims are sensitive to methodological details in data handling and evaluation. After correcting flaws like the "Drop Last Trick" in long-term forecasting, we found that ModernTCN, while competitive, is often outperformed by other leading models like PatchTST. Similarly, our anomaly detection experiments showed that its performance varies drastically; it lags behind simpler statistical methods on univariate data but excels in multivariate contexts. This highlights the importance of its architectural components, such as the cross-variable 'ConvFFN2' module, which our ablation study confirmed is crucial for handling missing data.

Furthermore, we addressed ModernTCN's oversight of a significant line of research in global convolutional models. Our analysis of effective receptive fields (ERFs) revealed that despite its claims of an enlarged ERF, it achieves a smaller receptive field compared to continuous kernel approaches like CCNN (Knigge et al., 2023), and at a much higher parameter cost. This finding motivated our main architectural innovation: integrating ModernTCN with a global continuous kernel for embedding irregularly sampled data. This hybrid model achieves new state-of-the-art performance on the PhysioNet 2019 dataset. This innovative combination opens new avenues for future research, particularly in exploring its application to other irregularly sampled datasets and those with extended sequence lengths.

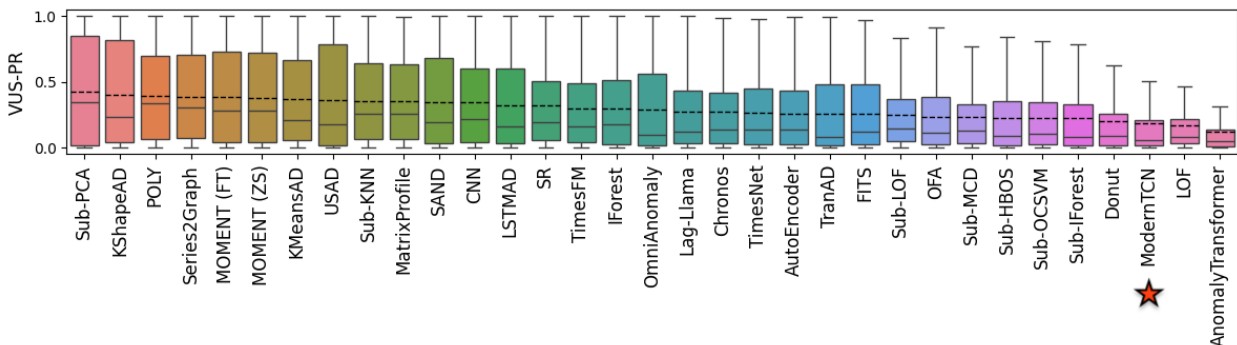

(a) Accuracy evaluation on univariate TSB-AD-U.

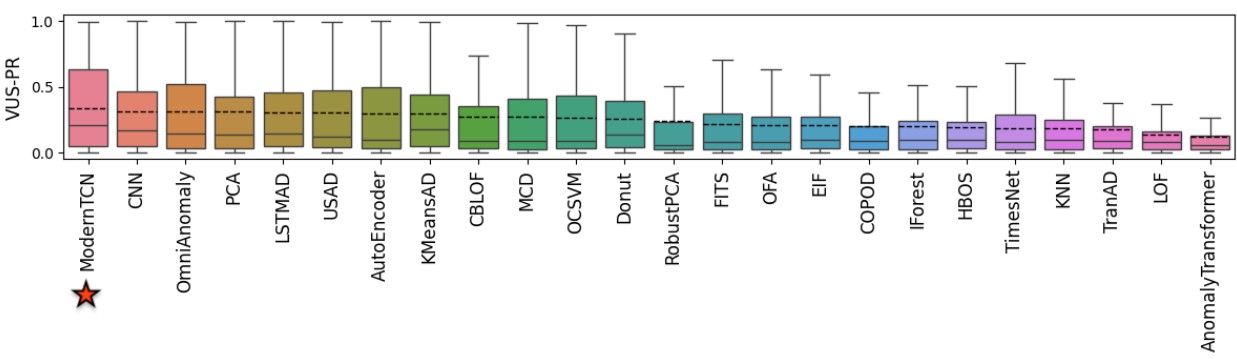

(b) Accuracy evaluation on multivariate TSB-AD-M.

Figure 3: Accuracy evaluation on univariate and multivariate time series anomaly detection. In the boxplot, the mean value is marked by a dashed line and the median by a solid line. Adapted from (Liu & Paparrizos, 2024).

Overall, this work highlights the importance of carefully considering experimental design choices in time series analysis. Some widely used practices—such as the "Drop Last Trick" in forecasting, using the test set for validation in classification and short-term forecasting, and point adjustment in anomaly detection—may unintentionally bias results. We encourage the community to carefully review these aspects and consider adopting benchmarking frameworks like the one we present here. Doing so can help ensure more robust, reliable, and comparable evaluations, ultimately fostering stronger progress in time series modeling.

### 5.1 Contact with the authors

The authors were contacted via email to request the code for ERF visualization and to discuss the flaws in the experimental setup. Unfortunately, no response was received.

### Acknowledgments

All experiments presented in this paper were run on the Dutch national supercomputer Snellius. We thank David W. Romero for his insightful comment on the OpenReview page of ModernTCN, which highlighted the overlooked research line of long convolutional models there and further motivated this study.

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

## Appendix

## Contents

**F   Short-Term Forecasting on ETT**                                                    **28**

# A    Experimental details

## A.1    Hyperparameters on the Extended Setups

For the original experiments, which aimed at reproducing results, we utilized the optimal settings reported in the ModernTCN paper. For the extended experiments, we conducted a grid search to determine the optimal hyperparameters. The grid for this search was constructed based on the optimal parameters of corresponding tasks as described in the ModernTCN paper. The specific details of the grid search and the selected parameters for the extended setups are provided in the tables below. Evaluations were performed on the validation set for 10 epochs, and the hyperparameters yielding the best validation score were selected.

Table 11: Hyperparameters for Speech Commands (MFCC)

| Parameter | Options Considered | Selected Value |
|---|---|---|
| Patch Size, Stride | 1,1; 8,4; 16,8 | 1,1 |
| Dimension | 32, 64, 128 | 32 |
| Kernel Size | 13, 31, 51, 71 | 13 |
| Learning Rate Scheduling | exponential (type1), step (type2), delayed exponential (type3) | type3 |
| Number of ModernTCN Blocks | 2, 3, 4 | 3 |

Table 12: Hyperparameters for Speech Commands (Raw)

| Parameter | Options Considered | Selected Value |
|---|---|---|
| Patch Size, Stride | 1,1; 16,8; 32,16; 64,32 | 32,16 |
| Dimension | 64, 128, 256, 512 | 512 |
| Kernel Size | 13, 31, 51, 71 | 71 |
| Learning Rate Scheduling | exponential (type1), step (type2), delayed exponential (type3) | type3 |
| Number of ModernTCN Blocks | 2, 3, 4 | 3 |

Table 13: Hyperparameters for PhysioNet

| Parameter | Options Considered | Selected Value |
|---|---|---|
| Patch Size, Stride | 1,1; 8,4; 16,8 | 1,1 |
| Dimension | 16, 32, 64, 128 | 32 |
| Kernel Size | 13, 31, 51, 71 | 31 |
| Learning Rate Scheduling | exponential (type1), step (type2), delayed exponential (type3) | type3 |
| Number of ModernTCN Blocks | 2, 3, 4 | 4 |

Table 14: Hyperparameters for Character Trajectories

| Parameter | Options Considered | Selected Value |
|---|---|---|
| Patch Size, Stride | 1,1; 8,4; 16,8 | 1,1 |
| Dimension | 16, 32, 64, 128 | 32 |
| Kernel Size | 13, 31, 51, 71 | 13 |
| Learning Rate Scheduling | exponential (type1), step (type2), delayed exponential (type3) | type3 |
| Number of ModernTCN Blocks | 2, 3, 4 | 2 |

Table 15: Hyperparameters for TSB-AD Anomaly Detection (Multivariate)

| Parameter | Options Considered | Selected Value |
|---|---|---|
| Window Size | 32, 96, 192 | 96 |
| Learning Rate | 1e-3, 1e-4, 1e-5 | 0.001 |
| Kernel Size | 13, 31, 51, 71 | 13 |

Table 16: Hyperparameters for TSB-AD Anomaly Detection (Univariate)

| Parameter | Options Considered | Selected Value |
|---|---|---|
| Window Size | 32, 96, 192 | 96 |
| Learning Rate | 1e-3, 1e-4, 1e-5 | 0.001 |
| Kernel Size | 13, 31, 51, 71 | 13 |

Table 17: Hyperparameters for Short Term Forecasting on ETTh1

| Parameter | Options Considered | Selected Value |
|---|---|---|
| Input Sequence Length (Pred Len 6) | 6x1, 6x2, 6x3 | 6x3 |
| Kernel Size (Pred Len 6) | 7, 13 | 13 |
| Input Sequence Length (Pred Len 12) | 12x1, 12x2, 12x3 | 12x3 |
| Kernel Size (Pred Len 12) | 7, 13, 31 | 13 |
| Input Sequence Length (Pred Len 18) | 18x1, 19x2, 18x3 | 18x3 |
| Kernel Size (Pred Len 18) | 7, 13, 31 | 13 |

Table 18: Hyperparameters for Short Term Forecasting on ETTm1

| Parameter | Options Considered | Selected Value |
|---|---|---|
| Input Sequence Length (Pred Len 6) | 6x1, 6x2, 6x3 | 6x3 |
| Kernel Size (Pred Len 6) | 7, 13 | 13 |
| Input Sequence Length (Pred Len 12) | 12x1, 12x2, 12x3 | 12x3 |
| Kernel Size (Pred Len 12) | 7, 13, 31 | 13 |
| Input Sequence Length (Pred Len 18) | 18x1, 19x2, 18x3 | 18x3 |
| Kernel Size (Pred Len 18) | 7, 13, 31 | 13 |

## A.2  Anomaly Detection Datasets of TSB-U-AD and TSB-M-AD

We extend the evaluation to TSB-AD (Liu & Paparrizos, 2024), a comprehensive benchmark for anomaly detection. This benchmark consists of 1070 time series from a diverse collection of 40 datasets, including 17 univariate and 13 multivariate datasets. For detailed characteristics, see Table 19.

Table 19: Summary characteristics of 40 datasets included in TSB-AD. '-' in the 2nd column indicates this dataset is transformed from the multivariate dataset. The 'Category' column indicates whether the datasets feature point anomalies (P) or sequence anomalies (Seq). Adapted from Liu & Paparrizos (2024)

| Name | # TS Collected | # TS Curated | Avg Dim | Avg TS Len | Avg # Anomaly | Avg Anomaly Len | Anomaly Ratio | Category |
|---|---|---|---|---|---|---|---|---|
| TSB-AD-U | | | | | | | | |
| UCR Wu & Keogh (2021) | 250 | 228 | 1 | 67818.7 | 1 | 198.9 | 0.6% | P&Seq |
| NAB Ahmad et al. (2017) | 58 | 28 | 1 | 5099.7 | 1.6 | 370.1 | 10.6% | Seq |
| YAHOO Laptev et al. (2015) | 367 | 259 | 1 | 1560.2 | 5.5 | 2.5 | 0.6% | P&Seq |
| IOPS IOP | 58 | 17 | 1 | 72792.3 | 25.6 | 48.7 | 1.3% | Seq |
| MGAB Thill et al. (2020) | 10 | 9 | 1 | 97777.8 | 9.7 | 20.0 | 0.2% | Seq |
| WSD Zhang et al. (2022) | 210 | 111 | 1 | 17444.5 | 5.1 | 25.4 | 0.6% | Seq |
| SED ? | 6 | 3 | 1 | 23332.3 | 14.7 | 64.0 | 4.1% | Seq |
| TODS Lai et al. (2021) | 15 | 15 | 1 | 5000.0 | 97.3 | 18.7 | 6.3% | P&Seq |
| NEK Si et al. (2024) | 48 | 9 | 1 | 1073.0 | 2.9 | 51.1 | 8.0% | P&Seq |
| Stock Tran et al. (2016) | 90 | 20 | 1 | 15000.0 | 1246.9 | 1.1 | 9.4% | P&Seq |
| Power Keogh et al. (2007) | 1 | 1 | 1 | 35040.0 | 4 | 750 | 8.5% | Seq |
| Daphnet (U) Bachlin et al. (2009) | - | 1 | 1 | 38774.0 | 6 | 384.3 | 5.9% | Seq |
| CATSv2 (U) Fleith (2023) | - | 1 | 1 | 300000.0 | 19.0 | 778.9 | 4.9% | Seq |
| SWaT (U) Mathur & Tippenhauer (2016) | - | 1 | 1 | 419919.0 | 27.0 | 1876.0 | 12.1% | Seq |
| LTDB (U) Goldberger et al. (2000) | - | 9 | 1 | 99700.0 | 127.5 | 144.5 | 18.6% | Seq |
| TAO (U) TAO | - | 3 | 1 | 10000.0 | 838.7 | 1.1 | 9.4% | P&Seq |
| Exathlon (U) Jacob et al. (2021) | - | 32 | 1 | 44075.4 | 3.1 | 1577.3 | 11.0% | Seq |
| MITDB (U) Goldberger et al. (2000) | - | 8 | 1 | 631250.0 | 68.7 | 451.9 | 4.2% | Seq |
| MSL (U) Hundman et al. (2018) | - | 9 | 1 | 3492.0 | 1.3 | 130.0 | 5.8% | Seq |
| SMAP (U) Hundman et al. (2018) | - | 19 | 1 | 7700.2 | 1.2 | 210.1 | 2.8% | Seq |
| SMD (U) Su et al. (2019a) | - | 38 | 1 | 24207.7 | 2.4 | 173.7 | 2.0% | Seq |
| SVDB (U) Greenwald (1990) | - | 20 | 1 | 171380.0 | 36.4 | 292.5 | 3.6% | Seq |
| OPP (U) Roggen et al. (2010) | - | 29 | 1 | 16544.8 | 1.4 | 653.4 | 6.4% | Seq |
| TSB-AD-M | | | | | | | | |
| GHL Filonov et al. (2016) | 48 | 25 | 19 | 199001.0 | 2.2 | 1035.2 | 1.1% | Seq |
| Daphnet Bachlin et al. (2009) | 17 | 1 | 9 | 38774.0 | 6.0 | 384.3 | 5.9% | Seq |
| Exathlon Jacob et al. (2021) | 72 | 27 | 21 | 60878.4 | 4.3 | 1373.3 | 9.8% | Seq |
| Genesis von Birgelen & Niggemann (2018) | 1 | 1 | 18 | 16220.0 | 3.0 | 16.7 | 0.3% | Seq |
| OPP Roggen et al. (2010) | 24 | 8 | 248 | 17426.75 | 1.4 | 394.3 | 4.1% | Seq |
| SMD Su et al. (2019a) | 28 | 22 | 38 | 25466.4 | 8.9 | 112.8 | 3.8% | Seq |
| SWaT Mathur & Tippenhauer (2016) | 4 | 2 | 59 | 207354.0 | 16.5 | 1093.6 | 12.7% | Seq |
| PSM Abdulaal et al. (2021) | 1 | 1 | 25 | 217624.0 | 72.0 | 338.6 | 11.2% | P&Seq |
| SMAP Hundman et al. (2018) | 54 | 27 | 25 | 7855.9 | 1.3 | 196.3 | 2.9% | Seq |
| MSL Hundman et al. (2018) | 27 | 16 | 55 | 3119.4 | 1.3 | 111.7 | 5.1% | Seq |
| CreditCard Sharafaldin et al. (2018) | 1 | 1 | 29 | 284807.0 | 465.0 | 1.1 | 0.2% | P&Seq |
| GECCO Moritz et al. (2018) | 1 | 1 | 9 | 138521.0 | 51.0 | 33.8 | 1.2% | Seq |
| MITDB Goldberger et al. (2000) | 48 | 13 | 2 | 336153.8 | 15.2 | 1846.8 | 2.7% | Seq |
| SVDB Greenwald (1990) | 78 | 31 | 2 | 207122.6 | 68.3 | 268.2 | 4.8% | Seq |
| LTDB Goldberger et al. (2000) | 7 | 5 | 2 | 100000.0 | 105.0 | 134.4 | 15.5% | Seq |
| CATSv2 Fleith (2023) | 10 | 6 | 17 | 240000.0 | 11.5 | 811.6 | 3.7% | Seq |
| TAO TAO | 45 | 13 | 3 | 10000.0 | 788.2 | 1.1 | 8.7% | P&Seq |

## A.3 TSB-AD Baselines

We compare ModernTCN against a wide range of baselines from the TSB-AD benchmark (Liu & Paparrizos, 2024), which are categorized into three groups:

- **Statistical Methods**: MCD (Rousseeuw & Driessen, 1999), OCSVM (Schölkopf et al., 1999), LOF (Breunig et al., 2000), KNN (Ramaswamy et al., 2000), KMeansAD (Yairi et al., 2001), CBLOF (He et al., 2003), POLY (Li et al., 2007), IForest (Liu et al., 2008), HBOS (Goldstein & Dengel, 2012), KShapeAD (Paparrizos & Gravano, 2015; 2017; Boniol et al., 2021), MatrixProfile (Yeh et al., 2016), PCA (Aggarwal, 2017), RobustPCA (Paffenroth et al., 2018), EIF (Hariri et al., 2019), SR (Ren et al., 2019), COPOD (Li et al., 2020), Series2Graph (Boniol & Palpanas, 2020), and SAND (Boniol et al., 2021).

- **Neural Network-based Methods**: AutoEncoder (Sakurada & Yairi, 2014), LSTMAD (Malhotra et al., 2015), Donut (Xu et al., 2018), CNN (Munir et al., 2018), OmniAnomaly (Su et al., 2019a), USAD (Audibert et al., 2020), AnomalyTransformer (Xu et al., 2021), TranAD (Tuli et al., 2022), TimesNet (Wu et al., 2023), and FITS (Xu et al., 2023).

- **Foundation Model-based Methods**: OFA (GPT4TS) (Zhou et al., 2023), Lag-Llama (Rasul et al., 2023), Chronos (Ansari et al., 2024), TimesFM (Das et al., 2024), and MOMENT (Goswami et al., 2024).

## A.4 Reproducibility and Baseline Sources

To reproduce the results presented in this paper, one can refer to the original repository. Some of the baseline results are produced in their official codebases; one can refer to the sources listed in Table 20.

---

[4]TiRex was run for 4 different input lengths, and the best results are reported in the main content. For detailed results, see Appendix C.4.

Table 20: Overview of Baseline Score Calculation Sources

| Task | Baseline Type / Source | Citation | Code Repository |
|---|---|---|---|
| Long-term Forecasting | Task-Specific Baselines | Qiu et al. (2024) | TFB |
| Long-term Forecasting | Foundational Models (excl. Tirex) | Li et al. (2025) | TSFM-Bench |
| Long-term Forecasting | Tirex | Auer et al. (2025) | Tirex (Submodule in RE-ModernTCN)[4] |
| Imputation | Baselines | Wu et al. (2023) | Time-Series-Library |
| Short-term Forecasting | Baselines | Wu et al. (2023) | Time-Series-Library |
| Speech Commands Classification | Baselines | - | CKConv, FlexConv, CCNN, S4 |
| Variable Length Time Series | Baselines | Lee & Shin (2025) | CKConv |
| Physionet 19 | Baselines | Romero et al. (2021b) | CKConv |
| Anomaly Detection | Baselines | Liu & Paparrizos (2024) | TSB-AD (Submodule in RE-ModernTCN) |

## B  Visualizing the ERF

Formally, let $I(n \times m \times l)$ be the input time series, where $n$ is the number of samples, $m$ is the number of variables, and $l$ is the input sequence length. Let $F(n \times m \times d \times l')$ be the final output feature map, we desire to measure the contributions of every time point on $I$ to the central points of every channel on $F$, *i.e.*, $F_{:,:,:,l'/2}$, which can be simply implemented via taking the derivatives of $F_{:,:,:,l'/2}$ to $I$ with the auto-grad mechanism. Concretely, we sum up the central points, take the derivatives to the input as the time-wise contribution scores and remove the negative parts (denoted by P). Then we aggregate the entries across all the examples and the input variables, and take the logarithm for better visualization. Formally, the aggregated contribution score matrix $A(l)$ is given by

$$P = \max\left(\frac{\partial(\sum_i^n \sum_j^m \sum_k^d F_{i,j,k,l'/2})}{\partial I}, 0\right), \tag{1}$$

$$A = \log_{10}\left(\sum_i^n \sum_j^m P_{i,j,:} + 1\right). \tag{2}$$

Then we respectively rescale A of each model to $[0, 1]$ via dividing the maximum entry for the comparability across models.

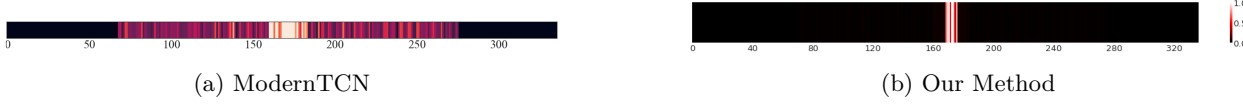

      (a) ModernTCN                     (b) Our Method

Figure 4: ERF visualizations from ModernTCN and our method for the same setup.

Following the mentioned methodology (from Kim et al. (2023) and Ding et al. (2022a)) as adapted in Luo & Wang (2024), we explored the ERF visualizations for the ETTh1 forecasting task with an input sequence length of 336. The ModernTCN approach claims to adapt these methodologies for 1D sequences; however, the specifics of their adaptation are not detailed, nor is the code available in their official repository. To establish a reference point, we visualized the same model under the same setup. As shown in Figure 4, the ERF visualization from ModernTCN exhibits more colored areas compared to our method. Such differences might be attributed to the color scheme used. In our visualizations, we clearly indicate the range of colors corresponding to values between $[0,1]$, whereas ModernTCN does not. Additionally, in our plots, not all black areas represent zero gradient contribution; rather, they are very close to zero relative to the maximum gradient contribution due to normalization. Upon closer inspection, the black regions near the center in our visualization reveal a slight red hue, indicating a minimal gradient contribution.

Nevertheless, as depicted in Figure 2, the global convolutions-based method demonstrates a significantly greater ability to enlarge the ERF compared to ModernTCN.

## C   Full Results

### C.1   ModernTCN Long-Term Forecasting

Table 21: Long-term forecasting results. "Reported": scores from ModernTCN paper. "Rerun": our reproduced results. "N.D.L.": No Drop Last, results with test loader not dropping incomplete last batches. "Len" indicates prediction length in time steps (96, 192, 336, and 720 for most datasets; 24, 36, 48, and 60 for ILI). Lower MSE/MAE indicates better performance. Values are multiplied by 1000 for brevity.

| Model | Len | ETTh1 MSE | ETTh1 MAE | ETTh2 MSE | ETTh2 MAE | ETTm1 MSE | ETTm1 MAE | ETTm2 MSE | ETTm2 MAE | Electricity MSE | Electricity MAE | Weather MSE | Weather MAE | Traffic MSE | Traffic MAE | Exchange MSE | Exchange MAE | ILI MSE | ILI MAE |
|---|---|---|---|---|---|---|---|---|---|---|---|---|---|---|---|---|---|---|---|
| Reported | 96 | 368 | 394 | 263 | 332 | 292 | 346 | 166 | 256 | 129 | 226 | 149 | 200 | 368 | 253 | 80 | 196 | 1347 | 717 |
|  | 192 | 405 | 413 | 320 | 374 | 332 | 368 | 222 | 293 | 143 | 239 | 196 | 245 | 379 | 261 | 166 | 288 | 1250 | 778 |
|  | 336 | 391 | 412 | 313 | 376 | 365 | 391 | 272 | 324 | 161 | 259 | 238 | 277 | 397 | 270 | 307 | 398 | 1388 | 781 |
|  | 720 | 450 | 461 | 393 | 433 | 416 | 417 | 351 | 381 | 191 | 286 | 314 | 334 | 440 | 296 | 656 | 582 | 1774 | 868 |
| Rerun | 96 | 369 | 394 | 264 | 333 | 297 | 348 | 169 | 256 | 131 | 227 | 150 | 204 | 371 | 256 | 81 | 197 | 1348 | 718 |
|  | 192 | 406 | 414 | 318 | 373 | 334 | 370 | 227 | 299 | 146 | 243 | 195 | 247 | 384 | 267 | 167 | 290 | 1448 | 820 |
|  | 336 | 392 | 412 | 314 | 376 | 371 | 395 | 276 | 329 | 166 | 264 | 237 | 283 | 404 | 273 | 314 | 402 | 1389 | 781 |
|  | 720 | 450 | 461 | 394 | 432 | 419 | 419 | 349 | 390 | 194 | 288 | 315 | 335 | 446 | 298 | 659 | 583 | 1775 | 868 |
| N.D.L. | 96 | 376 ± 2 | 397 ± 2 | 276 ± 1 | 341 ± 0 | 295 ± 1 | 346 ± 1 | 169 ± 2 | 256 ± 1 | 133 ± 0 | 228 ± 0 | 149 ± 1 | 203 ± 1 | 384 ± 1 | 270 ± 1 | 81 ± 0 | 198 ± 0 | 1935 ± 158 | 818 ± 38 |
|  | 192 | 410 ± 1 | 417 ± 1 | 338 ± 2 | 385 ± 1 | 333 ± 0 | 370 ± 1 | 225 ± 2 | 297 ± 2 | 149 ± 2 | 243 ± 3 | 195 ± 1 | 246 ± 1 | 395 ± 1 | 277 ± 1 | 168 ± 0 | 290 ± 0 | 2109 ± 140 | 951 ± 46 |
|  | 336 | 435 ± 1 | 433 ± 1 | 370 ± 3 | 414 ± 2 | 373 ± 2 | 395 ± 2 | 273 ± 1 | 328 ± 1 | 174 ± 2 | 269 ± 2 | 245 ± 1 | 284 ± 1 | 408 ± 1 | 284 ± 1 | 307 ± 0 | 399 ± 0 | 2008 ± 102 | 895 ± 25 |
|  | 720 | 456 ± 1 | 466 ± 1 | 410 ± 1 | 442 ± 0 | 418 ± 2 | 418 ± 1 | 350 ± 2 | 390 ± 2 | 201 ± 6 | 294 ± 4 | 316 ± 2 | 335 ± 1 | 453 ± 0 | 308 ± 1 | 814 ± 0 | 679 ± 0 | 1949 ± 93 | 904 ± 34 |

### C.2   ModernTCN Imputation

Table 22: Imputation task results with varying percentages of missing data (12.5%, 25%, 37.5%, 50%). "Original": reproduced results. "N.D.L.": No Drop Last, results with test loader not dropping incomplete last batches. "Miss %" indicates percentage of missing data. Lower MSE/MAE indicates better performance. All settings from original paper's repository. Values are multiplied by 1000 for brevity.

| Model | Miss % | ECL MSE | ECL MAE | Weather MSE | Weather MAE | ETTh1 MSE | ETTh1 MAE | ETTh2 MSE | ETTh2 MAE | ETTm1 MSE | ETTm1 MAE | ETTm2 MSE | ETTm2 MAE |
|---|---|---|---|---|---|---|---|---|---|---|---|---|---|
| Original | 12.5 | 59 | 170 | 24 | 39 | 34 | 126 | 38 | 122 | 16 | 83 | 18 | 77 |
|  | 25.0 | 69 | 181 | 25 | 41 | 44 | 143 | 40 | 127 | 18 | 89 | 19 | 80 |
|  | 37.5 | 81 | 196 | 27 | 44 | 54 | 157 | 43 | 134 | 22 | 96 | 21 | 85 |
|  | 50.0 | 84 | 199 | 31 | 52 | 68 | 175 | 48 | 142 | 27 | 106 | 23 | 91 |
| N.D.L. | 12.5 | 59 ± 1 | 170 ± 2 | 24 ± 0 | 40 ± 1 | 34 ± 0 | 126 ± 0 | 38 ± 0 | 122 ± 1 | 16 ± 0 | 83 ± 0 | 18 ± 0 | 76 ± 0 |
|  | 25.0 | 69 ± 1 | 181 ± 2 | 25 ± 0 | 41 ± 1 | 44 ± 0 | 143 ± 1 | 40 ± 0 | 127 ± 1 | 18 ± 0 | 89 ± 0 | 19 ± 0 | 80 ± 0 |
|  | 37.5 | 81 ± 1 | 196 ± 3 | 27 ± 0 | 44 ± 1 | 54 ± 1 | 157 ± 2 | 43 ± 1 | 134 ± 2 | 22 ± 0 | 96 ± 0 | 21 ± 0 | 85 ± 0 |
|  | 50.0 | 84 ± 1 | 199 ± 3 | 31 ± 0 | 52 ± 1 | 68 ± 1 | 175 ± 2 | 48 ± 1 | 142 ± 2 | 27 ± 0 | 106 ± 0 | 23 ± 0 | 91 ± 0 |

### C.3   Results on PhysioNet 2019 and Statistical Significance

This section provides a detailed breakdown of the model performance on the PhysioNet 2019 dataset, including the results from five separate runs for each model configuration. Table 23 presents the individual AUC scores, along with the mean and standard deviation, offering a comprehensive view of model stability and performance.

Table 23: Detailed AUC scores for models on the PhysioNet 2019 dataset over 5 runs.

| Model | Run 1 | Run 2 | Run 3 | Run 4 | Run 5 | Mean | Std Dev |
|---|---|---|---|---|---|---|---|
| CKConv | 0.868 | 0.888 | 0.884 | 0.897 | 0.876 | 0.883 | 0.011 |
| ModernTCN | 0.891 | 0.896 | 0.886 | 0.901 | 0.902 | 0.895 | 0.007 |
| ModernTCN - VarIndep | 0.880 | 0.869 | 0.875 | 0.873 | 0.876 | 0.875 | 0.004 |
| CKConv + ModernTCN | 0.901 | 0.902 | 0.901 | 0.908 | 0.908 | 0.904 | 0.004 |

The statistical significance of the performance improvement of the 'CKConv + ModernTCN' model over both 'CKConv' and 'ModernTCN' was evaluated using a one-sided Wilcoxon signed-rank test. This non-parametric test is suitable for comparing two related samples. In both comparisons, the alternative hypothesis

was that 'CKConv + ModernTCN' has a greater mean AUC score. The test yielded a p-value of 0.031 for both comparisons, indicating that the observed improvements are statistically significant.

The Python code snippet below, using the 'SciPy' library, was used to perform the statistical tests.

```python
from scipy.stats import wilcoxon

# AUC scores for CKConv, ModernTCN and CKConv + ModernTCN over 5 runs
ckconv_scores = [0.868, 0.888, 0.884, 0.897, 0.876]
moderntcn_scores = [0.891, 0.896, 0.886, 0.901, 0.902]
m_ckconv_scores = [0.901, 0.902, 0.901, 0.908, 0.908]

# One-sided Wilcoxon test: H1: M+CKConv > ModernTCN
stat_m, p_value_m = wilcoxon(m_ckconv_scores, moderntcn_scores, alternative='greater')
print(f"CKConv+ModernTCN vs ModernTCN: p-value={p_value_m:.3f}")

# One-sided Wilcoxon test: H1: M+CKConv > CKConv
stat_c, p_value_c = wilcoxon(m_ckconv_scores, ckconv_scores, alternative='greater')
print(f"CKConv+ModernTCN vs CKConv: p-value={p_value_c:.3f}")
```

### C.4 TiRex Full Results with Varying Input Lengths

TiRex was run with four different input lengths for each prediction length to identify the optimal configuration. The best-performing results are presented in the main long-term forecasting table. The full results, detailing performance across all tested input lengths, are provided below to show the sensitivity of the model to this hyperparameter.

Table 24: Performance with varying input (IL) and prediction (PL) lengths on long-term forecasting datasets (PL: 96, 192, 336, 720; IL: 336, 720, 1440, 2048). Note that the Illness dataset was evaluated with different prediction and input lengths (PL: 24, 36, 48, 60; IL: 36, 48, 60, 120), and these results are not included in this table.

| Configuration | ETTh1 MSE | ETTh1 MAE | ETTh2 MSE | ETTh2 MAE | ETTm1 MSE | ETTm1 MAE | ETTm2 MSE | ETTm2 MAE | Electricity MSE | Electricity MAE | Weather MSE | Weather MAE | Traffic MSE | Traffic MAE | Exchange MSE | Exchange MAE | Illness MSE | Illness MAE |
|---|---|---|---|---|---|---|---|---|---|---|---|---|---|---|---|---|---|---|
| **96(24)** | | | | | | | | | | | | | | | | | | |
| IL: 336 (36) | 0.412 | 0.387 | 0.306 | 0.333 | 0.341 | 0.338 | 0.197 | 0.257 | 0.126 | 0.212 | 0.173 | 0.204 | 0.411 | 0.250 | **0.093** | 0.210 | 5.827 | 1.463 |
| IL: 720 (48) | 0.382 | 0.377 | 0.298 | 0.327 | 0.323 | 0.329 | 0.188 | 0.248 | 0.121 | 0.207 | 0.161 | 0.192 | 0.386 | 0.240 | **0.093** | **0.209** | 5.138 | 1.445 |
| IL: 1440 (60) | **0.377** | **0.376** | 0.288 | 0.321 | 0.322 | 0.329 | 0.181 | 0.244 | **0.120** | **0.206** | 0.154 | 0.187 | 0.380 | 0.237 | 0.096 | 0.213 | 4.067 | 1.299 |
| IL: 2048 (120) | 0.377 | 0.377 | **0.274** | **0.315** | **0.304** | **0.323** | **0.174** | **0.239** | 0.121 | 0.207 | **0.151** | **0.183** | **0.378** | **0.235** | 0.097 | 0.213 | **1.493** | **0.744** |
| **192(36)** | | | | | | | | | | | | | | | | | | |
| IL: 336 (36) | 0.457 | 0.415 | 0.374 | 0.379 | 0.396 | 0.367 | 0.265 | 0.302 | 0.145 | 0.231 | 0.224 | 0.251 | 0.432 | 0.260 | **0.202** | **0.312** | 8.088 | 1.819 |
| IL: 720 (48) | 0.430 | 0.406 | 0.354 | 0.368 | 0.386 | 0.362 | 0.260 | 0.295 | 0.141 | 0.226 | 0.208 | 0.238 | 0.410 | 0.250 | 0.208 | 0.320 | 6.330 | 1.693 |
| IL: 1440 (60) | 0.425 | 0.406 | 0.347 | 0.364 | 0.379 | 0.360 | **0.247** | **0.289** | **0.140** | **0.225** | 0.200 | 0.232 | 0.405 | 0.247 | 0.216 | 0.325 | 4.835 | 1.476 |
| IL: 2048 (120) | 0.424 | 0.405 | **0.333** | **0.358** | **0.360** | **0.354** | 0.235 | **0.283** | 0.141 | 0.226 | **0.196** | **0.227** | **0.402** | **0.246** | 0.209 | 0.319 | **1.767** | **0.824** |
| **336(48)** | | | | | | | | | | | | | | | | | | |
| IL: 336 (36) | 0.485 | 0.430 | 0.373 | 0.389 | 0.435 | 0.389 | 0.313 | 0.333 | 0.164 | 0.250 | 0.282 | 0.291 | 0.444 | 0.267 | **0.382** | **0.441** | 8.247 | 1.889 |
| IL: 720 (48) | 0.464 | 0.425 | 0.355 | 0.379 | 0.431 | 0.386 | 0.317 | 0.333 | 0.159 | **0.244** | 0.266 | 0.280 | 0.425 | 0.258 | 0.410 | 0.459 | 6.247 | 1.717 |
| IL: 1440 (60) | **0.452** | 0.424 | 0.351 | 0.376 | 0.413 | 0.381 | 0.298 | 0.325 | **0.158** | **0.244** | 0.250 | 0.269 | 0.420 | 0.255 | 0.419 | 0.465 | 4.788 | 1.493 |
| IL: 2048 (120) | **0.452** | **0.422** | **0.348** | **0.375** | **0.396** | **0.376** | **0.285** | **0.318** | 0.159 | 0.245 | **0.246** | **0.265** | **0.418** | **0.254** | 0.387 | 0.445 | **1.814** | **0.845** |
| **720(60)** | | | | | | | | | | | | | | | | | | |
| IL: 336 (36) | 0.492 | 0.465 | 0.395 | 0.415 | 0.504 | 0.427 | 0.406 | 0.389 | 0.208 | 0.287 | 0.363 | 0.344 | 0.486 | 0.289 | **1.028** | **0.760** | 7.335 | 1.775 |
| IL: 720 (48) | **0.478** | **0.459** | 0.368 | 0.402 | 0.500 | 0.422 | 0.394 | 0.386 | 0.202 | **0.282** | 0.340 | 0.330 | 0.460 | 0.279 | 1.173 | 0.804 | 5.716 | 1.634 |
| IL: 1440 (60) | 0.488 | 0.465 | 0.366 | 0.402 | 0.462 | 0.411 | 0.365 | 0.373 | 0.201 | **0.282** | 0.316 | 0.316 | 0.454 | 0.276 | 1.133 | 0.794 | 4.599 | 1.458 |
| IL: 2048 (120) | 0.486 | 0.462 | **0.364** | **0.400** | **0.452** | **0.408** | **0.356** | **0.369** | **0.201** | **0.282** | 0.313 | 0.313 | 0.453 | 0.275 | 1.080 | 0.770 | **1.891** | **0.870** |

## D Drop Last Trick

The "Drop Last Trick" refers to discarding the last batch if it contains fewer instances than the batch size during evaluation. Table 25 examines how this affects model performance across different datasets. Our analysis reveals that significant changes in MSE and MAE primarily occur when more than 5% of the test set is dropped, as seen with ETTh1, ETTh2, Exchange (720), and ILI datasets. In most cases, performance metrics increase (positive $\Delta$ MSE/MAE values) when all data points are included, indicating worse performance as lower values are better for these metrics. This suggests that models may have been implicitly tuned for the smaller test sets. While most datasets show performance degradation when including all data points, cases like Exchange (336) show improvement, which aligns with the fact that dropping data points can lead to either better or worse performance depending on the characteristics of the excluded points. This indicates that this faulty setup significantly affects the results, especially when a high percentage of the test set is dropped due to high batch size and/or low test set size.

Table 25: Analysis of the "Drop Last Trick" impact across datasets. For each dataset, we show test set size, batch size, excluded data points, and performance differences. Numbers in parentheses indicate prediction length for forecasting tasks. Δ MSE/MAE shows the difference (No Drop Last - Rerun), with positive values indicating performance degradation when all data points are included.

| Task | Dataset | Test Set Size | Batch Size | Dropped Points | Dropped Points (%) | Δ MSE | Δ MAE |
|---|---|---|---|---|---|---|---|
| Long-term Forecasting | ETTm1 | 11,521 | 512 | 257 | 2.23% | +0.001 | 0.000 |
| Long-term Forecasting | ETTm2 | 11,521 | 512 | 257 | 2.23% | -0.001 | -0.001 |
| Long-term Forecasting | ETTh1 | 2,881 | 512 | 321 | 11.10% | +0.015 | +0.008 |
| Long-term Forecasting | ETTh2 | 2,881 | 512 | 321 | 11.10% | +0.025 | +0.015 |
| Long-term Forecasting | Electricity | 5,261 | 32 | 13 | 0.25% | +0.004 | +0.003 |
| Long-term Forecasting | Traffic | 3,509 | 32 | 21 | 0.60% | +0.009 | +0.006 |
| Long-term Forecasting | Weather (96) | 10,540 | 256 | 44 | 0.42% | -0.002 | -0.001 |
| Long-term Forecasting | Weather (192) | 10,540 | 256 | 44 | 0.42% | -0.001 | -0.001 |
| Long-term Forecasting | Weather (336) | 10,540 | 512 | 300 | 2.85% | +0.007 | +0.001 |
| Long-term Forecasting | Weather (720) | 10,540 | 512 | 300 | 2.85% | +0.003 | +0.001 |
| Long-term Forecasting | Exchange (96) | 1,422 | 128 | 14 | 0.98% | 0.000 | +0.001 |
| Long-term Forecasting | Exchange (192) | 1,422 | 128 | 14 | 0.98% | +0.001 | 0.000 |
| Long-term Forecasting | Exchange (336) | 1,422 | 512 | 398 | 27.99% | -0.007 | -0.003 |
| Long-term Forecasting | Exchange (720) | 1,422 | 512 | 398 | 27.99% | +0.155 | +0.096 |
| Long-term Forecasting | ILI | 170 | 32 | 10 | 5.88% | +0.510 | +0.095 |
| Imputation | ETTm1 | 11,521 | 16 | 1 | 0.01% | 0.000 | 0.000 |
| Imputation | ETTm2 | 11,521 | 16 | 1 | 0.01% | 0.000 | 0.000 |
| Imputation | ETTh1 | 2,881 | 16 | 1 | 0.03% | 0.000 | 0.000 |
| Imputation | ETTh2 | 2,881 | 16 | 1 | 0.03% | 0.000 | 0.000 |
| Imputation | Electricity | 5,261 | 16 | 13 | 0.25% | 0.000 | 0.000 |
| Imputation | Weather | 10,540 | 16 | 12 | 0.11% | 0.000 | 0.000 |

# E  Excluded Results

The results presented in this section were excluded from our main analysis due to methodological concerns regarding validation procedures. As detailed in the Experimental Setup section, both the M4 dataset for short-term forecasting and the UEA dataset for classification are provided with only train and test splits, with no designated validation set. Upon reviewing the source code, we identified that the test set was used for validation in both cases, introducing data leakage as the model gains indirect exposure to the test data during training through early stopping and hyperparameter selection.

A methodologically sound approach would involve creating a validation split from the training data. For instance, N-BEATS (Oreshkin et al., 2020) addresses this issue with M4 by first creating a validation split from the training set, finding optimal training settings on that validation set, and then training on the full training set using these optimal settings. However, we chose not to implement this approach for several reasons:

First, many of these datasets are already small for deep learning approaches. For example, M4-Weekly (359 samples), M4-Hourly (414 samples), and several UEA classification datasets like EthanolConcentration (261 samples) and Heartbeat (204 samples) have limited training samples to create a validation split from.

Second, implementing a proper validation approach like N-BEATS (Oreshkin et al., 2020) for ModernTCN would yield worse results (at best the same results) than the reported, as the training is on full training set in both cases. On top of that, those reported results are already worse than the reported results in N-BEATS Oreshkin et al. (2020).

Rather than allocating resources to rerun these experiments, we extended our study to new datasets such as multivariate short-term forecasting on ETT, time series classification on Speech Commands, and the PhysioNet 2019 sepsis challenge. These extensions allowed for more methodologically sound comparisons and a more comprehensive evaluation of ModernTCN's capabilities across diverse time series tasks.

The tables below present the reproduced results for completeness, but we emphasize that these should be interpreted with caution due to the methodological concerns outlined above.

## E.1  M4 Dataset Results

Table 26: Short-term forecasting task. Results are weighted averaged from several datasets under different sample intervals. Lower metrics indicate better performance. Reported refers to the scores reported in the ModernTCN paper. Rerun refers to the scores obtained by rerunning the model with the settings specified in the source code.

| Model | Yearly | | | Quarterly | | | Monthly | | | Others | | | Weighted Average | | |
|---|---|---|---|---|---|---|---|---|---|---|---|---|---|---|---|
| | SMAPE | MASE | OWA | SMAPE | MASE | OWA | SMAPE | MASE | OWA | SMAPE | MASE | OWA | SMAPE | MASE | OWA |
| Reported | 13.226 | 2.957 | 0.777 | 9.971 | 1.167 | 0.878 | 12.556 | 0.917 | 0.866 | 4.715 | 3.107 | 0.986 | 11.698 | 1.556 | 0.838 |
| Rerun | 13.231 | 2.957 | 0.777 | 10.001 | 1.170 | 0.881 | 12.598 | 0.920 | 0.868 | 4.835 | 3.175 | 1.009 | 11.732 | 1.561 | 0.841 |

## E.2 UEA Dataset Results

Table 27: Classification task. Accuracy metric is used. Reported refers to the scores reported in the ModernTCN paper. Rerun refers to the scores obtained by rerunning the model with the settings specified in the source code. Datasets: EthanolConcentration (EC), FaceDetection (FD), Handwriting (HW), Heartbeat (HB), JapaneseVowels (JV), PEMS-SF (PS), SelfRegulationSCP1 (SR1), SelfRegulationSCP2 (SR2), SpokenArabicDigits (SAD), UWaveGestureLibrary (UWL).

| | EC | FD | HW | HB | JV | PS | SR1 | SR2 | SAD | UWL | Averaged |
|---|---|---|---|---|---|---|---|---|---|---|---|
| Reported | 0.363 | 0.708 | 0.306 | 0.772 | 0.988 | 0.891 | 0.934 | 0.603 | 0.987 | 0.867 | 0.742 |
| Rerun | 0.319 | 0.687 | 0.284 | 0.771 | 0.981 | 0.832 | 0.928 | 0.617 | 0.981 | 0.859 | 0.726 |

# F Short-Term Forecasting on ETT

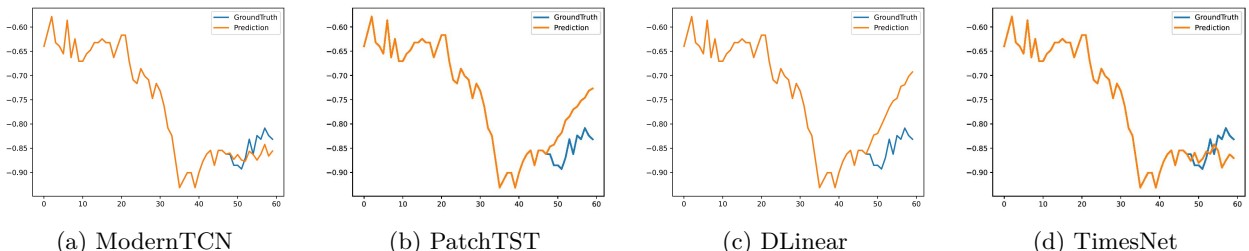

(a) ModernTCN     (b) PatchTST     (c) DLinear     (d) TimesNet

Figure 5: Short-term forecasting on ETTm1 with prediction length 12 and input length 48.

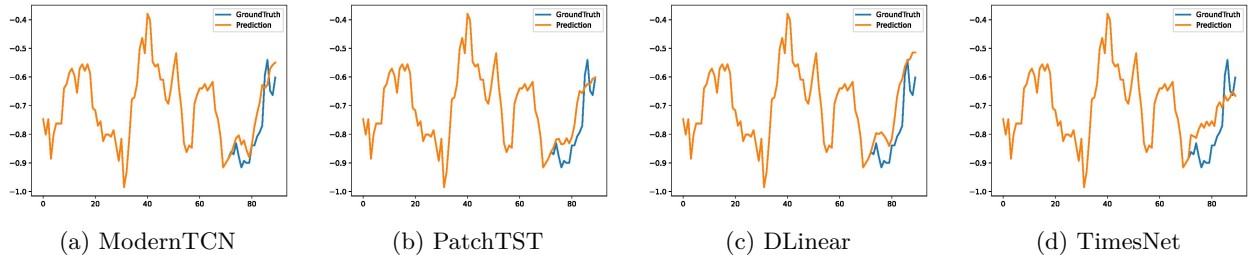

(a) ModernTCN      (b) PatchTST      (c) DLinear      (d) TimesNet

Figure 6: Short-term forecasting on ETTh1 with prediction length 18 and input length 72.

