# OpenReview forum: "ModernTCN Revisited: A Critical Look at the Experimental Setup in General Time Series Analysis"
_TMLR — Accepted by TMLR_

### Review · Reviewer_aWxW · 2025-03-28

**Summary Of Contributions:**

This study rigorously reproduces ModernTCN’s results across various time series tasks, extends the evaluation to additional datasets—including irregular and variable-length data—and identifies key methodological flaws, and offering new insights via effective receptive field (ERF) visualizations.

**Audience:**

Yes

**Broader Impact Concerns:**

No significant ethical or broader impact concerns are identified in this work.

**Claims And Evidence:**

Yes

**Requested Changes:**

I would suggest the followings to make the paper more robust and broadly applicable, thereby enhancing its contribution to the field of time series analysis.

## (Critical) Broaden the Scope and Refocus the Presentation:
The paper should be repositioned as a revisiting study of time series analysis in general, rather than solely focusing on ModernTCN. By broadening the perspective, the study would offer insights into common pitfalls—such as the "Drop Last Trick", data leakage, and threshold tuning—that can occur even in top-tier papers. The current inclusion of baselines like Transformer-based PatchTST indicates that a broader narrative is feasible, and rewriting the paper to address a wider range of time series models would significantly increase its general interest and impact.

## (Less Critical) Statistical Reporting:
Although the experimental evaluation is extensive, incorporating error bars or statistical significance measures for the results would further strengthen the claims. For example, reporting confidence intervals or standard deviations for performance metrics such as MSE or MAE would add rigor. Given the breadth of experiments, we acknowledge that this may be challenging, but even a subset of statistically robust comparisons would enhance the credibility of the findings.

## (Least Critical) Extend the Study to Variable-Length Time Series:
The current study largely focuses on fixed-length time series. Evaluating additional TCN variants that are designed to handle variable-length inputs (e.g., [3-6]), as well as benchmarking against other architectures such as state-space models and Transformers, would provide a more comprehensive overview of the field. This extension is essential to determine whether the issues identified—like the sensitivity to experimental design—are unique to ModernTCN or represent broader challenges across different time series modeling approaches.

[3] E. J. Keogh and C. A. Ratanamahatana. Exact indexing of dynamic time warping. Knowledge and Information Systems, 2002.
[4] M. Schneider, et al. An end-to-end machine learning approach with explanation for time series with varying lengths. Neural Comput. Appl., 2024.
[5] A. Sawada, et al. Convolutional neural networks for time-dependent classification of variable-length time series. 2022 International Joint Conference on Neural Networks (IJCNN), 2022.
[6]  H. Lee and D. Shin. Beyond information distortion: Imaging variable-length time series data for classification. Sensors, 2025.

**Strengths And Weaknesses:**

# Strengths
## Reproducibility Commitment:
The authors provide a comprehensive re-run study of ModernTCN with detailed experimental setups and code disclosures in the Appendix. This commitment to reproducible research is commendable and crucial given the current reproducibility challenges in machine learning.

## Identification of Experimental Pitfalls:
The paper exposes several key issues in experimental design, such as the "Drop Last Trick", using test data for validation, and tuning thresholds on the test data. These insights are valuable for the community and offer actionable guidelines for improving experimental rigor.

## Thorough Empirical Evaluation:
The authors conduct extensive experiments across multiple time series tasks (e.g., long-term forecasting, imputation, anomaly detection, and classification) and include advanced analyses such as ERF visualizations. These methodologies enhance our understanding of the underlying model behavior.


# Weaknesses

## Narrow Focus on ModernTCN:
While revisiting studies are essential, the paper primarily focuses on the ModernTCN architecture. Although ModernTCN currently represents state-of-the-art performance in certain tasks, its heuristic nature and lack of theoretical guarantees (unlike approaches based on, e.g., Rough Path theory [1, 2]) may limit the long-term impact of the findings. By concentrating almost exclusively on one neural network structure, the paper's relevance might diminish as more advanced models emerge.

[1] T. Lyons, et al. Differential equations driven by rough paths. 2007.
[2] T. Lyons and A. D. McLeod. Signature methods in machine learning. arXiv, 2022.

---

> ### Author Response · Authors · 2025-04-18
> **Response to Reviewer 2**
>
> We sincerely thank the reviewer for their thorough feedback which has helped improve our study significantly. We address each point:
>
> **Regarding Weakness - Narrow Focus:**
>
> We have broadened the focus of our study to validate the common general time series analysis setup that originates from TimesNet [1], which is widely adopted in the literature (e.g., TimeMixer++, ICLR 2025 Oral - https://openreview.net/forum?id=1CLzLXSFNn ). We have added relevant baselines for each task that were missing in the earlier version, making the study more comprehensive and impactful for the broader time series analysis community.
>
> **Regarding Requested Changes:**
>
> 1. **Broaden Scope and Refocus:** We have repositioned the study to examine common experimental setups in time series analysis, particularly focusing on methodological issues that persist even in recent publications. We've expanded our analysis to include various baselines (transformer-based, MLP-based, and convolution-based). Additionally, we adapted a comprehensive anomaly detection setup/benchmark as an extension (https://thedatumorg.github.io/TSB-AD/).
>
> 2. **Statistical Reporting:** We now report results over 5 runs, including both mean and standard deviation. For cases where results are averaged over different setups (e.g., long-term forecasting with different prediction lengths or imputation with different masking ratios), we omit standard deviation as it would not be informative due to the natural variation between different setups.
>
> 3. **Variable Length Time Series:** We have added a variable length time series classification task using the character trajectories dataset. We compare our results with the two best-performing baselines from [Lee and Shin, 2025] using both accuracy and macro average precision metrics, providing a comprehensive evaluation of performance on variable-length inputs.
>
> [1] Haixu Wu, Tengge Hu, Yong Liu, Hang Zhou, Jianmin Wang, and Mingsheng Long. Timesnet:
> Temporal 2d-variation modeling for general time series analysis. ICLR,
> 2023.

---

### Review · Reviewer_NpRp · 2025-04-05

**Summary Of Contributions:**

This paper is a reproducibility study of the ModernTCN model, a recently proposed convolutional network architecture designed for general time series analysis. ModernTCN aims to improve Temporal Convolutional Networks (TCNs) by using large kernel sizes to increase the effective receptive field (ERF), enabling the capture of long-term dependencies. The authors of this study validate ModernTCN’s claims regarding performance across multiple tasks such as forecasting, imputation, classification, and anomaly detection. They also extend the evaluation to additional datasets (Speech Commands and PhysioNet), compare ModernTCN against other convolutional architectures (CKConv, FlexConv, CCNN, and S4), and investigate experimental methodologies and components like the cross-variable module through ablation studies. Overall, the paper finds that ModernTCN achieves competitive performance but identifies several limitations related to experimental sensitivity, parameter efficiency, ERF effectiveness, and anomaly detection evaluation.

**Audience:**

Yes

**Claims And Evidence:**

Yes

**Requested Changes:**

1. Clearly report results both with and without the tricks (Section 4.1.1). A detailed analysis of why the model’s performance deteriorates significantly without it would be helpful.
2. Provide clear details on anomaly detection threshold selection and discuss the implication of “point adjustment” evaluation (Section 4.1.3). Additionally, explicitly compare the performance against straightforward baseline models to validate the effectiveness of ModernTCN.
3.  Include and discuss effective receptive field visualization comparisons explicitly (as shown in Figure 2). Clarify the reason behind the observed discrepancy between the claimed large ERF and the visualized results.
4. Add explicit discussions on ModernTCN’s parameter count relative to competing convolutional models and the implications on computational resources and training time (Table 7)
5. Clarify and justify kernel size selection for classification tasks (Tables 6 / 8). Explain why performance does not follow clear trends across kernel sizes, and provide guidelines or recommendations for kernel size choice based on experimental results.
6. Clearly document library and framework versions to ensure reproducibility is robust to minor implementation variations noted in the rerun results (Section 4.1).
7. Provide a brief analysis or insight into why ModernTCN underperforms convolutional models with global receptive fields on datasets such as Speech Commands, despite employing large kernels (Table 7).
8. In the ablation study on ConvFFN2 (Section 4.2.3), clarify explicitly how and why the cross-variable dependencies improve model handling of missing values and irregular sampling

While I enjoy the topic to systematically reproduce a series of methods, I feel it requires more analysis on the pros / cons and why it succeeds or fails in certain cases.

**Strengths And Weaknesses:**

Strenths:
1. The paper thoroughly reproduces the ModernTCN experiments and provides additional evaluations on new tasks and datasets.
2. It expands the original paper’s scope by including comparisons against additional convolution-based models with global receptive fields (e.g., CKConv, S4, CCNN), providing a broader context.
3. Highlights significant issues such as the “Drop Last Trick,” data leakage, and anomaly detection evaluation flaws, which are often overlooked but impactful.
4. Clearly demonstrates the role of the cross-variable component (ConvFFN2) in handling missing values on irregularly sampled data.

Weaknesses:
1. ModernTCN’s performance significantly deteriorates when methodological issues such as the “Drop Last Trick” are addressed (Section 4.1.1), suggesting sensitivity to experimental setup, which is not analyzed thoroughly
2. Compared to convolution-based models with global receptive fields (e.g., CCNN and S4), ModernTCN uses considerably more parameters yet achieves lower performance on certain classification tasks, notably Speech Commands (Section 4.2.2, Table 7).
3. Despite claims of improved receptive fields, visualization results show ModernTCN’s ERF is relatively limited compared to CCNN, questioning the effectiveness of the model’s large kernels
4. The anomaly detection protocol (“point adjustment”) severely overestimates model effectiveness, as a naive baseline outperforms ModernTCN (Table 4).

---

> ### Author Response · Authors · 2025-04-18
> **Response to Reviewer 3**
>
> We sincerely thank the reviewer for their detailed feedback. We address each point raised:
>
> **Regarding Weaknesses:**
>
> 1. The effect of methodological issues like the "Drop Last Trick" is now thoroughly analyzed in the main content and further detailed in Appendix D.
>
> 2-4. These points highlight limitations of the ModernTCN method itself rather than our reproducibility study. While we don't directly address these methodological weaknesses, they are related to the requested changes which we address below.
>
> **Regarding Requested Changes:**
>
> 1. Results with and without the tricks are now clearly reported in Tables 2 and 4. A detailed analysis of performance deterioration without the "Drop Last Trick" is provided in Appendix D.
>
> 2. The anomaly detection threshold calculation methodology is now detailed in Section 3.3.1. We have adopted the TSB-AD benchmark (https://thedatumorg.github.io/TSB-AD/), which includes statistical methods, neural networks, and foundational methods, providing a more comprehensive evaluation.
>
> 3. ERF visualization results are now explicitly discussed at the end of Section 4.2.2 and Appendix B, including analysis of the discrepancy between claimed and visualized ERF results.
>
> 4. Parameter count comparisons with competing convolutional models are added to Section 4.2.2. Claims about time efficiency have been removed as they were insufficiently substantiated.
>
> 5. Instead of adapting recommended hyperparameters, we now conduct comprehensive hyperparameter tuning (detailed in Appendix A.1). Tables 6 and 8 have been replaced with comparisons of the best-performing ModernTCN variants against other baselines.
>
> 6. Library and framework versions are documented in the code provided in the supplementary material.
>
> 7. Initial underperformance on Speech Commands was due to using recommended hyperparameters. After comprehensive tuning and architectural adjustments (downsampling the signal with high stride and increasing model dimension for expressivity), particularly addressing the challenging 16,000 sequence length, we achieved improved results.
>
> 8. Analysis of cross-variable interactions is expanded in Sections 4.1.2 and 4.2.4, demonstrating how ModernTCN leverages inter-variate dependencies for handling missing values and irregularly sampled data. The impact of the cross-variable component (ConvFFN2) is evidenced by performance degradation when removed, as shown in the "ModernTCN - Variable Independent" variant.

---

### Review · Reviewer_Yx9H · 2025-04-05

**Summary Of Contributions:**

This paper systematically reproduces and expands ModernTCN, verifies its performance on a variety of time series tasks, and finds its advantages and disadvantages in different scenarios. By evaluating ModernTCN on more datasets and tasks, and providing an effective visualization implementation of the receptive field (ERF), it helps to understand the characteristics of the model.

**Audience:**

No

**Claims And Evidence:**

No

**Requested Changes:**

+ 1. Add more evaluations and discussions of TCN-Based baselines.

**Strengths And Weaknesses:**

> Strength:

+ 1. The original experimental settings were carefully examined during the reproduction process, and modified comparative experiments were conducted.

+ 2. The ERF visualization tool code of ModernTCN was provided to the community (but it does not seem to come from the author), which promoted the further development of related research.

> Weakness:

+ 1. This work is unsuitable for TMLR. Even as a reproducibility study, its scope is too narrow, focusing solely on ModernTCN code with limited community impact.

+ 2. Innovation is limited. As a reproducibility paper, its novelty lies only in experimental extensions and deeper analysis, not in new methodologies. The identified issues, like the "Drop Last Trick," have been previously noted in benchmarking papers.

+ 3. It offers no effective suggestions for improvement. For instance, when ModernTCN underperforms on tasks suited to convolutional methods with global receptive fields (e.g., Speech Commands classification) and shows low parameter efficiency, the authors provide no better alternatives.

+ 4. Despite detecting some methodological issues, there's still room for improvement in experimental setups for certain tasks. The use of "point adjustment" protocols in anomaly detection tasks is one such area where refinement is needed.

---

> ### Author Response · Authors · 2025-04-18
> **Response to Reviewer 1**
>
> We sincerely thank the reviewer for their thorough evaluation and constructive feedback, which has helped us significantly improve our study. We address each point raised:
>
> **Regarding the Weaknesses:**
>
> 1. **Scope and Impact:** We have broadened the focus of our study to validate the common general time series setup that originates from TimesNet, which is widely adopted in the literature (e.g., TimeMixer++, ICLR 2025 Oral - https://openreview.net/forum?id=1CLzLXSFNn). We have added relevant baselines for each task that were missing in the earlier version, making the study more comprehensive and impactful for the broader time series analysis community.
>
> 2. **Innovation:** Our study introduces an architectural innovation by combining ModernTCN with CKConv, achieving state-of-the-art performance on PhysioNet 2019. While issues like the "Drop Last Trick" have been previously noted, their persistence in recent works (including ICLR 2025 submissions) underscores the importance of addressing these methodological concerns. Additionally, to the best of our knowledge, the usage of test set for validation on M4 and 10 selected time series classification datasets has not been previously highlighted in the literature.
>
> 3. **Improvement Suggestions:** For Speech Commands (raw), we introduced effective improvements by downsampling the signal during embedding with a high stride value and increasing the model dimension to 512 for enhanced expressivity. These modifications resulted in ModernTCN variants outperforming CKConv. Additionally, our comprehensive hyperparameter tuning led to improved performance across extended datasets.
>
> 4. **Experimental Setup:** To address the point adjustment issue in anomaly detection, we adopted the TSB-AD benchmark (https://thedatumorg.github.io/TSB-AD/), a comprehensive framework that directly addresses this concern and provides a more robust evaluation methodology.
>
> **Regarding the Requested Changes:**
>
> As suggested, we have expanded our evaluations to include more TCN-based baselines and added a related work section that discusses various TCN and convolution-based methods, providing a more thorough context for our analysis.

---

> > ### Comment · Reviewer_Yx9H · 2025-04-20
> > **Thanks to authors**
> >
> > I appreciate the hard work of the authors, and they have addressed most of my concerns appropriately. However, my current concerns regarding weakness 1 remain, which I consider to be the main shortcoming of this paper.

---

### Decision · Action_Editor_osfH · 2025-05-20

**Recommendation:** Accept with minor revision

**Comment:**

This paper provides a well-executed reproducibility study of ModernTCN, offering significant methodological improvements and addressing several important issues in time series analysis. While the authors have made improvements in the breadth of their study, further revision is needed to strengthen the generalizability and impact of their findings.

Strengths:
- The paper demonstrates strong reproducibility efforts and provides valuable insights on ModernTCN, including performance analysis across various tasks and datasets.
- It identifies and addresses methodological issues like the "Drop Last Trick" and the use of test data for validation, which contribute to the broader understanding of time series analysis models.
- The addition of relevant baselines and the extension of evaluations to new tasks make the study more comprehensive.

Weaknesses:
- The focus on ModernTCN limits the paper's long-term impact, and the framing of the paper still heavily revolves around this model. A more general presentation would enhance the paper’s relevance.
- While the paper addresses methodological flaws, it lacks clear suggestions for improving ModernTCN’s performance on tasks where it underperforms compared to models with global receptive fields (e.g., Speech Commands).
- There is room for improvement in experimental setups, particularly for tasks like anomaly detection and classification with varying input lengths.

Revision Requests:
- Broaden the scope of the paper to include more time series models and not focus exclusively on ModernTCN, which has been addressed by the authors
- Include more detailed analysis and justification for certain methodological choices, such as kernel size selection and parameter tuning.
- Provide statistical significance for the experimental results and include error bars where possible to strengthen the claims made.

**Audience:**

Yes, the findings are of interest to the TMLR audience, particularly in the field of time series analysis and reproducibility studies. The paper contributes valuable insights into the performance of ModernTCN and its limitations, addressing methodological flaws and offering improvements to experimental setups.

**Claims And Evidence:**

Yes, the claims are supported by accurate and clear evidence. The authors have demonstrated thorough experimentation and analysis, providing convincing evidence through additional baselines, methodological analyses, ablation studies, and new insights.

---

> ### Author Response · Authors · 2025-05-25
> **Clarification on "Broaden the scope" revision point**
>
> Dear Action Editor,
>
> Thank you for the "Accept with minor revision" decision and your feedback on our submission.
>
> We are preparing the camera-ready version and would appreciate a small clarification on one of the revision requests:
>
> Regarding the point to "Broaden the scope of the paper," we noted your comment "has been addressed by the authors." In our last revision, we did expand the scope (e.g., by repositioning the study and adding further baselines). Could you please confirm if these changes in our latest revision are what you refer to as having addressed this point, thereby satisfying this request already? Otherwise, further clarification would be very much appreciated.

---

> ### Author Response · Authors · 2025-06-19
> **Submission of the Camera-Ready Version**
>
> Dear Action Editor,
>
> Thank you for the decision and helpful feedback. We appreciate your guidance throughout the review process.
>
> We have submitted the **camera-ready version**, with the following updates made in response to the revision requests:
>
> ---
>
> **Changes Since Last Submission:**
>
> - **Broadened Scope of the Paper:**
>   Added 6 foundational time series forecasting baselines, including recent SOTA models such as TiRex [1]. Each experiment now includes at least 3 baselines to ensure broader coverage and more meaningful comparisons.
>
> - **Methodological Justifications:**
>   Provided detailed kernel size selection and hyperparameter tuning strategy in Section 3.3 and Appendix A.1.
>
> - **Statistical Significance and Error Bars:**
>   Included a one-sided Wilcoxon signed-rank test for PhysioNet 2019 results (confirming the significance of SOTA results).
>   Added standard deviations over 5 runs for forecasting and imputation tasks as well (Appendices C.1 and C.2).
>
> - **Refocused Paper Objective:**
>   Revised the main content, abstract, introduction, discussion, and title to reflect the paper's broader objective of auditing the standard general time series analysis setup beyond ModernTCN.
>
> ---
>
> Thanks again to you and the reviewers for the valuable feedback—we believe that the reviews significantly improved the shape and clarity of the paper.
>
>
> Best regards,
>
>
> [1] Andreas Auer, Patrick Podest, Daniel Klotz, Sebastian Böck, Günter Klambauer, and Sepp Hochreiter.
> **TiRex: Zero-Shot Forecasting Across Long and Short Horizons with Enhanced In-Context Learning.**
> *arXiv:2505.23719*, 2025.